# Supraphysiological activation of TAK1 promotes skeletal muscle growth and mitigates neurogenic atrophy

Anirban Roy [1] & Ashok Kumar [1✉]

Skeletal muscle mass is regulated through coordinated activation of multiple signaling pathways. TAK1 signalosome has been found to be activated in various conditions of muscle atrophy and hypertrophy. However, the role and mechanisms by which TAK1 regulates skeletal muscle mass remain less understood. Here, we demonstrate that supraphysiological activation of TAK1 in skeletal muscle of adult mice stimulates translational machinery, protein synthesis, and myofiber growth. TAK1 causes phosphorylation of elongation initiation factor 4E (eIF4E) independent of mTOR. Inactivation of TAK1 disrupts neuromuscular junction morphology and causes deregulation of Smad signaling. Using genetic approaches, we demonstrate that TAK1 prevents excessive loss of muscle mass during denervation. TAK1 favors the nuclear translocation of Smad4 and cytoplasmic retention of Smad6. TAK1 is also required for the phosphorylation of eIF4E in denervated skeletal muscle. Collectively, our results demonstrate that TAK1 supports skeletal muscle growth and prevents neurogenic muscle atrophy in adult mice.

[1] Department of Pharmacological and Pharmaceutical Sciences, University of Houston College of Pharmacy, Houston, TX, USA.
✉email: akumar43@Central.UH.EDU

Skeletal muscle is the most abundant tissue in human body that provides structural support and allows mobility as well as serves as a storage site for glucose, lipids, and protein to be utilized for energy production during catabolic periods. Skeletal muscle mass is governed through coordinated activation of catabolic and anabolic pathways[1,2]. The IGF1/Akt/mTOR axis has been proposed as a major signaling pathway that stimulates protein synthesis resulting in skeletal muscle growth[1]. Activation of this pathway also inhibits muscle atrophy through suppressing the expression of various atrogenes[3–5]. Intriguingly, more recent studies have suggested that the constitutive activation of mTORC1 causes skeletal muscle atrophy and myopathies without having any significant effect on protein synthesis. For example, genetic ablation of mTORC1 inhibitor TSC1 (tuberous sclerosis complex 1) exacerbates denervation-induced atrophy in mice. The denervation-induced increase in mTOR inhibits Akt kinase through a feedback mechanism, which, in turn, results in the activation of Forkhead Box subfamily O (FoxO) transcription factors and increased gene expression of E3 ubiquitin ligases, MAFbx and MuRF1[6]. Moreover, sustained activation of mTORC1 inhibits autophagy in skeletal muscle leading to the onset of myopathies at later stages[7]. In addition, acute activation or suppression of mTORC1 has been found to dysregulate autophagy and impair muscle mass homeostasis in denervated muscle[8]. Additionally, mTORC1 contributes to age-related muscle atrophy potentially through enhancing the phosphorylation of STAT3 and increasing gene expression of GDF15 (growth differentiation factor 15)[9].

Members of transforming growth factor-β (TGF-β) superfamily, which comprises the activin/myostatin/TGF-β subfamily and the bone morphogenetic protein (BMP) subfamily, also play an important role in the regulation of skeletal muscle mass in various conditions[1,2]. Binding of activin/myostatin/TGF-β group of ligands to activin type IIB and type IIA receptors (ActRIIB/IIA) and TGF-β receptors (TGFβRII) leads to recruitment and activation of type 1 receptor, activin receptor-like kinase (ALK)-4, -7, and -5. ALK4/7 and ALK5 phosphorylate Smad2/3 to promote formation of a multiprotein complex with co-Smad, Smad4. The Smad2/3-Smad4 complex then translocates to the nucleus to regulate expression of Smad2/3 target genes such as *Nodal*, *Pitx2*, *Lefty1* and *Lefty2*[1,10]. The BMP/GDF subfamily ligands preferentially bind to a combination of type II receptors that includes BMP type II receptor (BMPRII), ActRIIA, and ActRIIB which then leads to the recruitment of type I receptors, ALK3, ALK6, and ALK2. This complex promotes phosphorylation and interaction of Smad1/5/8 with Smad4 to regulate nuclear gene expression. While Smad4 is a co-activator, Smad6 and Smad7 negatively regulate the activity of Smad2/3 and Smad1/5/8[1,10,11].

The Smad2/3 complex activated by activin/myostatin/TGF-β subfamily of ligands inhibits skeletal muscle growth and promotes skeletal muscle wasting through augmenting the expression of MAFbx and MuRF1, and repressing protein synthesis[1,10,11]. Indeed, inhibition of Smad2/3-mediated signaling using genetic or pharmacological approaches has been found to prevent skeletal muscle wasting in catabolic states and in various genetic muscle disorders[12,13]. Recent studies have provided strong evidence that BMP-Smad1/5/8 signaling axis positively regulates skeletal muscle mass[11,13–15]. Forced activation of this pathway through overexpression of BMP7 or constitutively active ALK3 mutant is sufficient to induce skeletal muscle growth and to mitigate the loss of skeletal muscle mass after denervation. By contrast, the absence of GDF5/BMP14 or Smad6-mediated blockade of BMP-Smad1/5/8 axis abolishes myofiber hypertrophy in myostatin-deficient mice and exacerbates denervation-induced muscle atrophy in mice[11,13,14]. Therefore, maintenance and growth of skeletal muscle mass require a synchronized balance between Smad2/3 and Smad1/5/8 signaling.

TGF-β-activated kinase 1 (TAK1), originally identified as a member of the MAPK kinase kinase (MAP3K) family, is a major component of the non-canonical TGF-β signaling pathway[16,17]. For activation, TAK1 forms a complex with adaptor protein TAB1 and either TAB2 or TAB3 and undergoes a conformational change that leads to the autophosphorylation at Thr-187 and Ser-192 in the activation loop[18,19]. TAK1 phosphorylates and activates several signaling cascades, including p38 MAPK and JNK. TAK1 also phosphorylates I kappa B kinase β (IKKβ) that leads to the activation of nuclear factor-kappa B (NF-κB) transcription factor[18,20]. TAK1 has also been found to regulate canonical TGF-β signaling in some cell types. For example, TAK1 is required for the BMP receptor-mediated activation of Smad1/5/8 and expression of BMP target genes in chondrocytes[21]. We previously demonstrated that TAK1 is expressed in satellite stem cells and is essential for regeneration of skeletal muscle in adult mice[22]. Moreover, we have demonstrated that TAK1 is required for the maintenance of skeletal muscle mass in adult mice. Targeted inactivation of TAK1 leads to severe muscle wasting which is accompanied with increased proteasome activity, heightened autophagy, redox imbalance, and mitochondrial dysfunction[23,24]. However, the role of TAK1 in the regulation of various signaling pathways during muscle growth or atrophy remains unknown. Furthermore, the impact of forced activation of TAK1 in the regulation of skeletal muscle mass has not been yet investigated.

In this study, using genetic mouse models and adeno-associated virus serotype 6 (AAV6) vectors, we demonstrate that supraphysiological activation of TAK1 through overexpression of TAK1 and TAB1 protein induces muscle hypertrophy and protein synthesis. Moreover, our results show that inactivation of *Tak1* gene leads to neurogenic atrophy associated with degeneration of neuromuscular junction (NMJ). Forced activation of TAK1 attenuates denervation-induced muscle atrophy in adult mice. Our study has also identified a novel interplay between TAK1 and Smad1 during neurogenic muscle atrophy.

## Results

**Forced activation of TAK1 promotes skeletal muscle hypertrophy in mice.** We have previously reported that targeted inducible inactivation of TAK1 leads to skeletal muscle wasting in adult mice[23,24]. However, it remains unknown how TAK1 is regulated in skeletal muscle in response to anabolic stimuli and whether the activation of TAK1 is sufficient to promote muscle growth. We used the bilateral synergistic ablation (SA) model that has been widely used to induce hypertrophy of the plantaris muscle[25–27]. In this model, removal of the lower portion of the gastrocnemius and soleus muscle places a functional overload on the remaining plantaris muscle during normal mouse movement, which results in muscle hypertrophy. Our analysis showed that there was significant increase in the levels of phosphorylated TAK1 (p-TAK1) protein in plantaris muscle of wild-type mice after 14d of SA surgery. We also found a significant up-regulation in the levels of phosphorylated p38 (p-p38), phosphorylated Smad1/5/8 (p-Smad1/5/8), phosphorylated Smad2 (p-Smad2), and phosphorylated mTOR (p-mTOR) proteins in plantaris muscle at day 14 after SA surgery suggesting that TAK1 is activated in association with other established muscle growth regulatory proteins. We also found that total levels of TAK1, Smad1, and Smad2 protein were also considerably increased in plantaris muscle in response to functional overload (Fig. 1a, b).

The N-terminal kinase domain of TAK1 forms a complex with TAB1 which is essential for TAK1 auto-phosphorylation at S192 and its activation[28,29]. Previous studies have shown that TAK1 has no kinase activity when ectopically expressed alone but is activated when co-expressed with TAB1[19,30]. To understand the

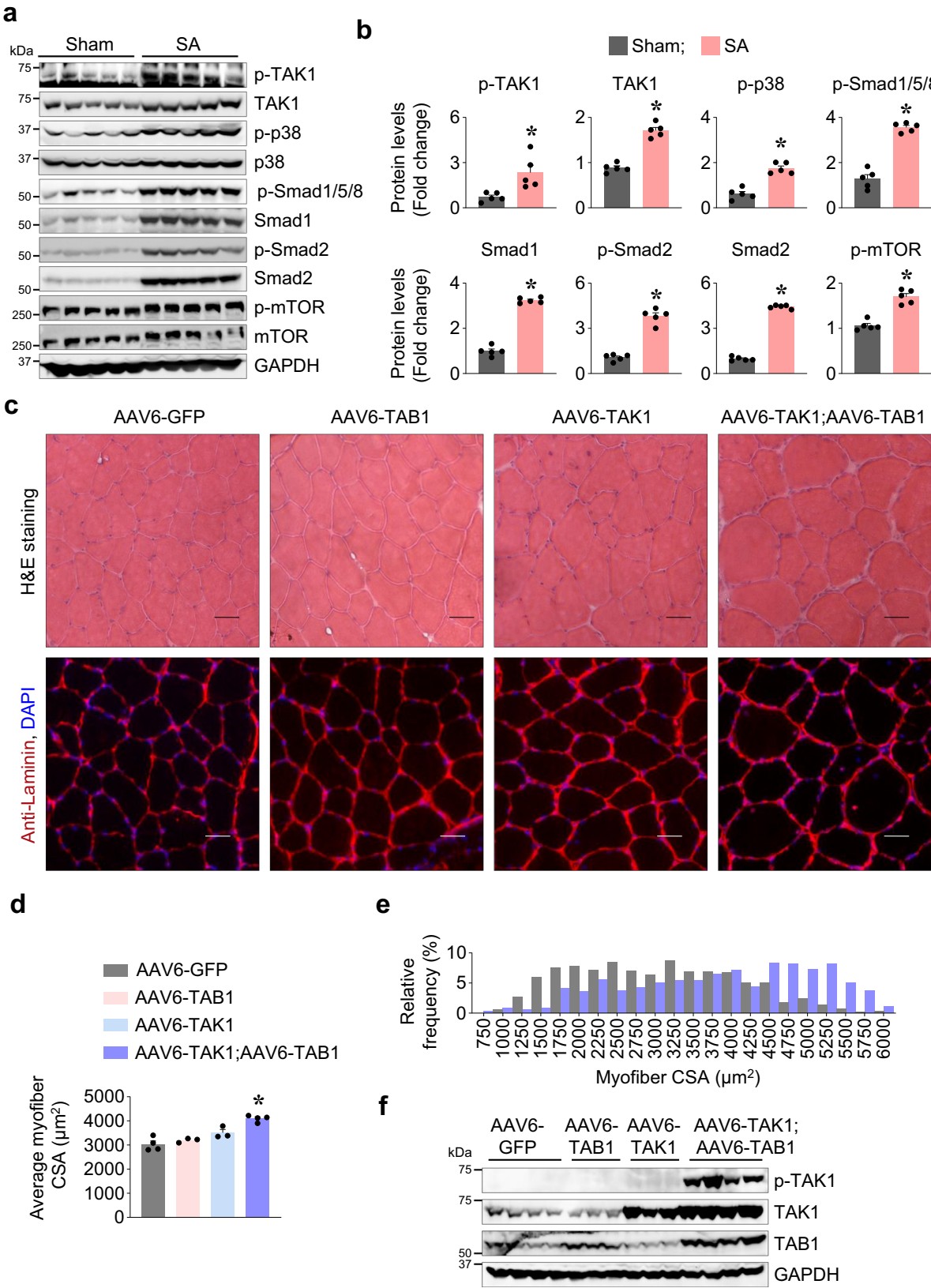

role of TAK1 in the regulation of muscle mass, we used AAV serotype 6 (AAV6) vectors having a constitutive cytomegalovirus (CMV) promoter to overexpress TAK1, TAB1, or green fluorescence protein (GFP) in skeletal muscle of mice. Our initial studies showed that intramuscular injection of about 1–2 × 10$^{10}$ AAVs per TA muscle of adult mice results in GFP expression in most of the myofibers at day 5 (Supplementary Fig. 1a). Therefore, we transduced left side TA muscle of wild-type mice with 30 μl PBS solution containing AAV6-TAB1 (2.5 × 10$^{10}$ vg), AAV6-TAK1 (2.5 × 10$^{10}$ vg), or a combination of AAV6-TAB1 (1.25 × 10$^{10}$ vg) and AAV6-TAK1 (1.25 × 10$^{10}$ vg) while the contralateral right TA muscle was injected with AAV6-GFP

**Fig. 1 Forced activation of TAK1 promotes myofiber hypertrophy.** C57BL/6 wild type mice were subjected to sham or bilateral synergistic ablation (SA) surgery. After 14 days, plantaris muscle was isolated and analyzed by western blot. **a** Immunoblots, and **b** densitometry quantification demonstrating levels of p-TAK1, TAK1, p-p38, p-38, p-Smad1/5/8, Smad1, p-Smad2, Smad2, p-mTOR, mTOR and GAPDH protein in control and overloaded plantaris muscle of mice ($n = 5$). Data presented as mean ± SEM. *$p < 0.05$, values significantly different from corresponding sham-operated TA muscle by unpaired t-test. Left side TA muscle of C57BL/6 mice was given intramuscular injection of AAV6-TAB1 ($2.5 \times 10^{10}$ vg), AAV6-TAK1 ($2.5 \times 10^{10}$ vg), or a combination of AAV6-TAB1 ($1.25 \times 10^{10}$ vg) and AAV6-TAK1 ($1.25 \times 10^{10}$ vg) while the contralateral right TA muscle was injected with AAV6-GFP ($2.5 \times 10^{10}$ vg) particles. After 28 days, the mice were euthanized and the TA muscles were harvested and analyzed. **c** Representative photomicrographs of TA muscle transverse sections after H&E staining or anti-laminin and DAPI staining. Scale bar, 50 μm. **d** Average myofiber cross-sectional area (CSA) in TA muscle of mice injected with AAV6-GFP, AAV6-TAK1, AAV6-TAB1, or a combination of AAV6-TAK1 and AAV6-TAB1. **e** Relative frequency distribution of myofiber CSA in TA muscle injected with AAV6-GFP or combination of AAV6-TAK1 and AAV6-TAB1. $n = 3-4$ mice in each group. **f** Immunoblots presented here demonstrate levels of p-TAK1 and total TAK1, TAB1, and GAPDH protein in TA muscle of mice after 28 days of intramuscular injection of AAV6-GFP, AAV6-TAB1, AAV6-TAK1, or combination of AAV6-TAB1 and AAV6-TAK1. Data presented as mean ± SEM and was analyzed by one-way ANOVA followed by Tukey's multiple comparison test. *$p < 0.05$, values significantly different from the corresponding AAV6-GFP injected TA muscle.

($2.5 \times 10^{10}$ vg). After 28 days, the mice were euthanized, and the TA muscle was isolated and analyzed by morphometric and biochemical assays. There was no significant difference in wet weight of TA muscle (normalized with body weight) injected with AAV6-TAK1 or AAV6-TAB1 alone compared to contralateral TA muscle injected with AAV6-GFP. Interestingly, there was a significant increase in the wet weight of TA muscle injected with both AAV6-TAK1 and AAV6-TAB1 compared to contralateral TA muscle injected with AAV6-GFP (Supplementary Fig. 1b). Hematoxylin & Eosin (H&E) staining showed that AAV6-mediated overexpression of GFP, TAB1, TAK1, or TAK1 and TAB1 together did not produce any overt phenotype in TA muscle of mice (Fig. 1c). Immunostaining of TA muscle sections for laminin protein followed by quantitative analysis showed that the average myofiber CSA was significantly higher in TA muscle injected with AAV6-TAK1 and AAV6-TAB1 compared to TA muscle injected with AAV6-TAK1, AAV6-TAB1 or AAV6-GFP alone (Fig. 1c, d). Furthermore, our analysis showed that the proportion of myofibers with higher cross-sectional area (CSA) was considerably increased only in TA muscle co-injected with AAV6-TAK1 and AAV6-TAB1 compared to TA muscle injected with AAV6-GFP alone (Fig. 1e). There was no major effect on relative frequency distribution of myofiber CSA in TA muscle injected with AAV6-TAK1 or AAV6-TAB1 compared with AAV6-GFP injected control (Supplementary Fig. 1c, d). Consistent with published reports[28,29], we found that the phosphorylation of TAK1 was drastically increased in TA muscle overexpressing both TAK1 and TAB1, but not TAK1 or TAB1 alone (Fig. 1f). In addition, Trichrome Masson's staining of TA muscle sections showed that forced activation of TAK1 through intramuscular injection of AAV6-TAK1 and AAV6-TAB1 does not cause inflammation or fibrosis in skeletal muscle (Supplementary Fig. 1e). Similar to TA muscle, we found that intramuscular co-injection of AAV6-TAK1 and AAV6-TAB1 significantly increased average myofiber CSA, frequency of myofibers with higher CSA, and activation of TAK1 in GA muscle of wild-type mice (Supplementary Fig. 2a-d). Collectively, these results suggest that TAK1 is activated during overload-induced muscle growth and forced activation of TAK1 through overexpression of TAK1 and TAB1 is sufficient to promote myofiber hypertrophy in adult mice.

**TAK1 stimulates protein translational machinery in skeletal muscle of mice.** To understand the potential mechanisms by which activation of TAK1 promotes myofiber hypertrophy, we measured the phosphorylation of various proteins that regulate skeletal muscle mass. TAK1 is known to activate canonical NF-κB and MAPK signaling pathways in mammalian cells[20]. Consistently, we found that the levels of phosphorylated p65 (p-p65),

phosphorylated ERK1/2 (p-ERK1/2), and phosphorylated (p-p38) MAPK were significantly increased in TA muscle overexpressing TAK1 and TAB1 compared to contralateral TA muscle expressing GFP (Fig. 2a, b). We next measured the levels of phosphorylated and total Smad proteins. A significant increase in the total Smad1 and Smad2 levels, no difference in p-Smad1/5/8 and Smad4 levels, and a minor reduction in p-Smad2 level was observed in TA muscle of mice overexpressing TAK1 and TAB1 compared to contralateral control muscle. By contrast, there was no significant difference in the phosphorylated or total levels of Akt, mTOR, or AMPK protein between control and TAK1 with TAB1 over-expressing TA muscle (Fig. 2a, b). Western blot analysis confirmed that the levels of phosphorylated as well as total TAK1 protein were significantly increased in TA muscle of mice injected with AAV6-TAK1 and AAV6-TAB1 (Fig. 2a, b).

We next examined the phosphorylation of key factors that regulate protein translation during skeletal muscle growth. Protein synthesis in mammalian cells is dependent on multiple eukaryotic translation initiation factors (eIFs) that initiate translation[31,32]. Western blot analysis revealed a significant increase in the levels of phosphorylated eIF4E (p-eIF4E), phosphorylated eIF4B (p-eIF4B) and total eIF4H, but not total eIF4E, total eIF4B or eIF4A in TA muscle overexpressing TAK1 and TAB1 compared to contralateral control muscle expressing GFP alone (Fig. 2c, d). eIF2 is another essential factor for forming the 43 S pre-initiation complex, a requisite for protein synthesis. Activation of eIF2 is negatively regulated by phosphorylation of α subunit at Ser51 residue[33,34]. We found that the levels of phosphorylated eIF2α protein were significantly reduced in TA muscle overexpressing TAK1 and TAB1 protein compared with contralateral control muscle. In response to various anabolic stimuli, mTOR phosphorylates p70S6 kinase (p70S6K) which in turn phosphorylates the ribosomal protein S6 (rpS6) to positively affect muscle growth[35]. While we found significant increase in the levels of phosphorylated rpS6 (p-rpS6) protein, the levels of phosphorylated p70S6K (p-p70S6K) remained comparable between control and TAK1 with TAB1 overexpressing TA muscle (Fig. 2c, d). In response to mitogenic and/or stress stimuli, p38 MAPK phosphorylate and activate MAPK interacting protein kinase 1 and 2 (Mnk1 and 2) which then phosphorylate their major downstream effector, the eIF4E at Ser209 to initiate protein translation[36]. In addition, ERK1/2 activate p90RSK which then phosphorylates rpS6 at Ser235/236 in vitro and in vivo using an mTOR-independent mechanism to stimulate Cap-dependent initiation of translation[37]. Interestingly, we found that levels of phosphorylated Mnk1 (p-Mnk1) as well as phosphorylated p90RSK (p-p90RSK) were significantly increased in TA muscle overexpressing TAK1 and TAB1 compared to contralateral TA muscle expressing GFP only (Fig. 2c, d).

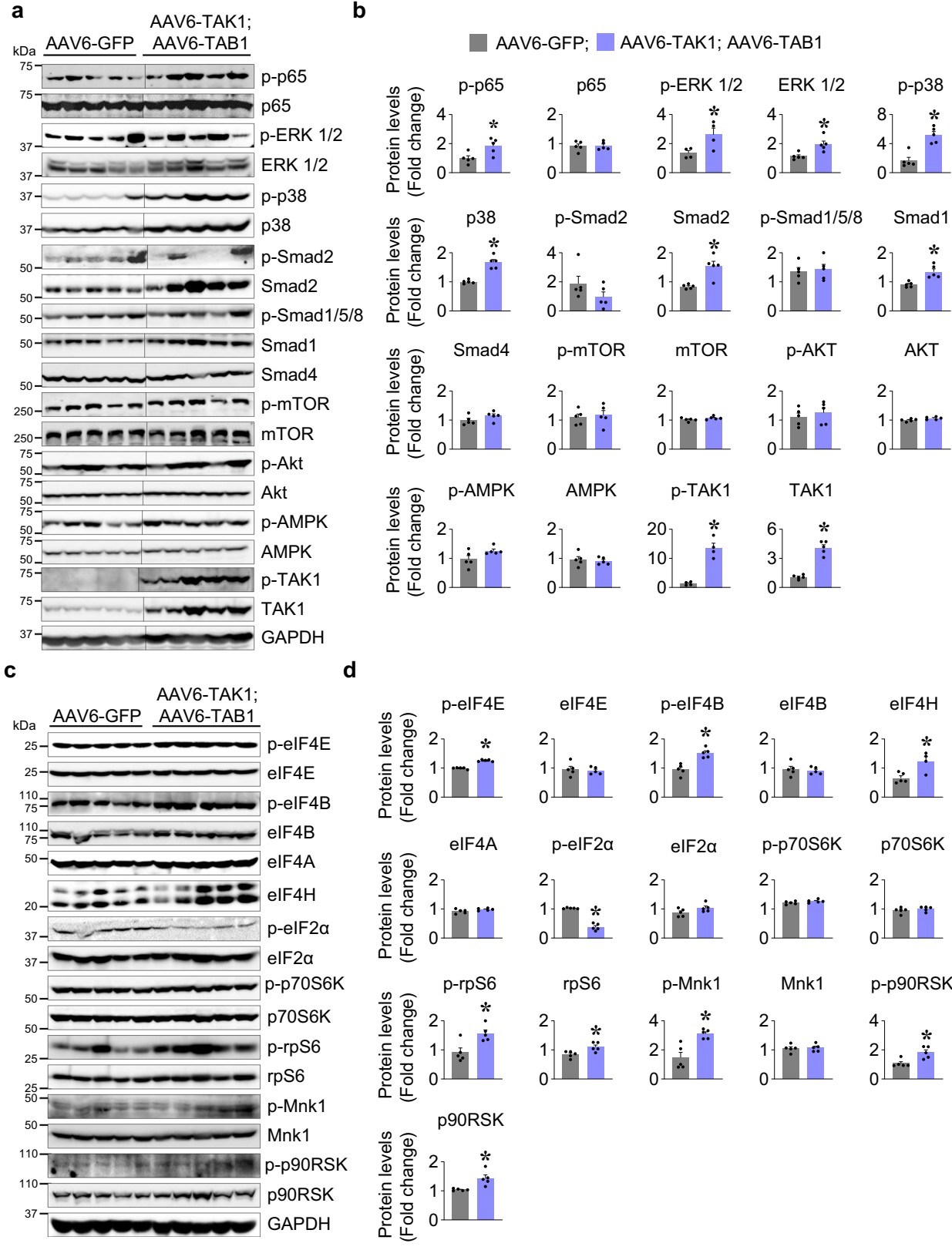

By performing surface sensing of translation (SUnSET) assay, we also measured the rate of protein synthesis in TA muscle of mice overexpressing GFP, TAB1, TAK1, or a combination of TAK1 and TAB1. Results showed that there was a significant increase in protein synthesis in TA muscle overexpressing TAK1 and TAB1 compared to those expressing GFP, TAB1, or TAK1 alone (Supplementary Fig. 3a, b). Our results also confirmed that the levels of phosphorylated eIF4E and rpS6 are considerably increased in TA muscle of mice co-expressing TAK1 and TAB1, but not TAB1 or TAK1 alone (Supplementary Fig. 3c, d).

**Fig. 2 TAK1 stimulates protein translational machinery in skeletal muscle of mice. a** Representative immunoblots and **b** densitometry analysis showing levels of p-p65, p65, p-ERK1/2, ERK1/2, p-p38, p38, p-Smad2, Smad2, p-Smad1/5/8, Smad1, Smad4, p-mTOR, mTOR, p-AKT, AKT, p-AMPK, AMPK, p-TAK1, TAK1 and GAPDH protein in TA muscle at day 28 after intramuscular injection of AAV6-GFP or equal amounts of AAV6-TAK1 and AAV6-TAB1. GAPDH was used as loading control. **c** Representative immunoblots and **d** densitometry analysis showing levels of p-eIF4E, eIF4E, p-eIF4B, eIF4B, eIF4A, eIF4H, p-eIF2α, eIF2α, p-p70S6K, p70S6K, p-rpS6, rpS6, p-Mnk1, Mnk1, p-p90RSK, p90RSK, and GAPDH in TA muscle at day 28 after intramuscular injection of AAV6-GFP or a combination of AAV6-TAK1 and AAV6-TAB1. Black vertical lines on the immunoblots indicate that intervening lanes have been spliced out. $n = 5$ mice in each group. Data presented as mean ± SEM. *$p < 0.05$, values significantly different from corresponding AAV6-GFP injected muscle by unpaired t-test.

**Forced activation of TAK1 stimulates protein synthesis in cultured myotubes.** We also studied the effect of overexpression of TAK1 and TAB1 on the activation of various signaling proteins in cultured myotubes. Primary myoblasts prepared from wild-type mice were differentiated into myotubes by incubation in differentiation medium for 72 h. The myotubes were then transduced with AAV6-GFP or a combination of AAV6-TAK1 and AAV6-TAB1 for 48 h. Myotubes transduced with AAV6-TAK1 and AAV6-TAB1 or AAV6-GFP alone appeared healthy (Fig. 3a). Consistent with in vivo results, there was a significant increase in the levels of p-rpS6, p-eIF4E, eIF4E, p-Mnk1 ($P = 0.07$), Mnk1, p-ERK1/2, p-p90RSK, and Smad1 in TAK1- and TAB1-overexpressing cultures compared to control cultures (Fig. 3b, c). By contrast, there was no significant difference in the levels p-mTOR and p-Smad1/5/8 between control and TAK1- and TAB1-overexpressing cultures (Fig. 3b, c). We also performed SUnSET assay to measure the rate of protein synthesis. Results showed that the rate of protein synthesis was significantly increased in cultures transduced with AAV6-TAK1 and AAV6-TAB1 compared to control cultures transduced with AAV6-GFP alone (Fig. 3d, e).

Since forced activation of TAK1 induces protein synthesis without affecting the phosphorylation of mTOR, we next sought to investigate whether TAK1 can rescue protein synthesis in cultured myotubes upon inhibition of mTOR. Cultured myotubes were co-transduced with AAV6-TAK1 and AAV6-TAB1 or with AAV6-GFP for 48 h followed by treatment with rapamycin for an additional 24 h. Results showed that rapamycin inhibited the levels of phosphorylated rpS6 protein and the rate of protein synthesis in both GFP- and TAK1- with TAB1-overexpressing cultures. Interestingly, the rate of protein synthesis was significantly higher in rapamycin-treated TAK1- and TAB1-overexpressing cultures compared to corresponding rapamycin-treated control cultures expressing GFP alone (Fig. 3f, g). Taken together, these results suggest that forced activation of TAK1 stimulates the translational machinery independent of mTOR.

**Inactivation of TAK1 leads to NMJ derangements and muscle atrophy.** Structural changes along with functional alterations result in neuromuscular junction (NMJ) impairment in many neurodegenerative disorders and aging[38]. We have previously reported that targeted inducible inactivation of TAK1 in adult mice leads to severe muscle wasting and kyphosis which are reminiscent of age-related muscle wasting[23]. Using muscle-specific tamoxifen inducible TAK1-knockout (Tak1mKO) mice, we investigated whether TAK1 has any role in maintaining the integrity of NMJs in mice. Littermate floxed Tak1 (Tak1fl/fl) and Tak1mKO (Tak1fl/fl; HSA-MCM) mice were treated with tamoxifen for 4 consecutive days. Thereafter, the mice were fed with a tamoxifen-containing chow for next 24 days. Finally, the mice were euthanized, and individual hind-limb muscle was isolated and analyzed by morphometric and biochemical methods. Consistent with our previously published reports[23,24], there was a substantial reduction in average myofiber CSA in Tak1mKO mice compared with Tak1fl/fl mice (Supplementary Fig. 4a, b).

The vertebrate NMJ is composed of the presynaptic nerve terminal, intra synaptic basal lamina (synaptic cleft), post-synaptic specialization (motor end plate of muscle fiber), and presynaptic Schwann cells[39]. We examined the impact of muscle-specific TAK1 inactivation on NMJ morphology. Longitudinal sections generated from TA muscle of Tak1fl/fl and Tak1mKO mice were stained with α-bungarotoxin conjugated Alexa 488, antibodies against neurofilament (NF-M)) and synaptic vesicle protein 2 (SV2), and DAPI (Fig. 4a). The NMJ components were analyzed according to Image-J based standardized methodology, NMJ-morph[40]. We observed a gross derangement at the post-synaptic region of NMJ in Tak1mKO mice. There was a significant decrease in AChR area and end plate area in the NMJs of Tak1mKO mice compared to the littermate Tak1fl/fl mice (Fig. 4b). The NMJs of Tak1mKO mice had significantly lesser synaptic overlap, area of synaptic contact and compactness when compared to NMJs of Tak1fl/fl mice (Fig. 4b). Moreover, there was increased fragmentation of NMJs in Tak1mKO mice compared with Tak1fl/fl mice (Fig. 4c). Some of the endplates in the Tak1mKO mice were oddly smaller whereas some were larger but with extensive fragmentation (Supplementary Fig. 4c, d). By contrast, the parameters of the pre-synaptic region such as axonal diameter, nerve terminal area, average length of branches, and complexity index remained comparable between Tak1fl/fl and Tak1mKO mice (Supplementary Fig. 4e).

Several studies have suggested that nerve injury, denervation, or synaptic blockade leads to increased synthesis of acetylcholine receptors (AChRs) which is associated with high turnover of AChRs and NMJ instability[41–43]. Therefore, we measured the gene expression of various components of NMJ in skeletal muscle of Tak1fl/fl and Tak1mKO mice. Interestingly, the mRNA levels of AChRα (Chrna1), AChRβ (Chrnb1), Agrin (Agrn), MuSK (Musk) and Dok7 were significantly upregulated in GA muscle of Tak1mKO mice compared to Tak1fl/fl mice (Fig. 4d) confirming that inactivation of TAK1 in skeletal muscle leads to the disruption of NMJs. In a separate experiment, we also measured the mRNA levels of various components of NMJ in control and denervated GA muscle of mice. Results showed a similar upregulation in the mRNA levels of AChRα (Chrna1), AChRβ (Chrnb1) and Musk in denervated GA muscle compared to contralateral sham-operated muscle of wild-type mice (Supplementary Fig. 5a).

FOXO family of transcription factors, such as FoxO1, 3, and 4, have been found to induce gene expression of various components of ubiquitin-proteasome system (UPS) and autophagy[44]. Overexpression of FoxO proteins induces muscle atrophy whereas their inhibition prevents myofiber atrophy in multiple conditions, including denervation[1,2,44]. We observed a significant increase in the levels of FoxO3 and FoxO4, but not FoxO1 protein, in GA muscle of Tak1mKO mice compared to Tak1fl/fl mice (Fig. 4e, f). Next, we examined the HDAC4-myogenin axis which is another major pathway triggering proteolysis in denervated muscle[1,2]. This pathway is also involved in synaptogenesis and indirectly suppresses the expression of molecules involved in glycolysis and corepressors of myogenin[45,46]. Results showed that the mRNA and protein levels of HDAC4 and myogenin (Myog) were significantly increased in GA muscle of Tak1mKO mice compared to Tak1fl/fl

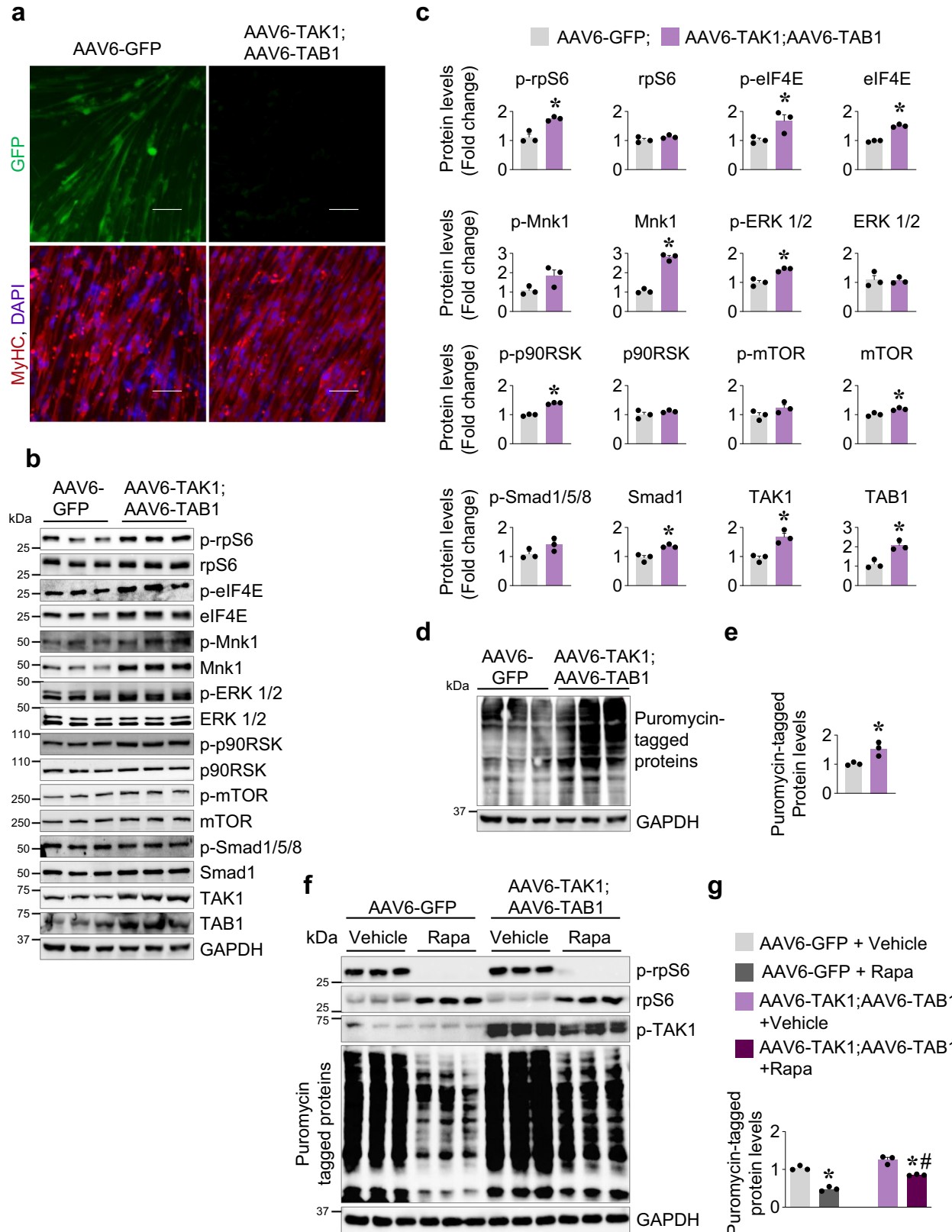

mice (Fig. 4g, h and i). These results are consistent with our findings in wild-type mice where similar upregulated expression of *Hdac4* and *Myog* was observed in GA muscle in response to sciatic nerve transection (Supplementary Fig. 5b). Collectively, these results suggest that TAK1 plays a pivotal role in maintaining NMJ homoeostasis in adult mice.

**Inactivation of Tak1 disrupts Smad signaling in skeletal muscle.** Accumulating evidence suggests that Smad signaling plays an important role in the regulation of skeletal muscle mass in the conditions of muscle growth and atrophy[1,2]. Negative growth regulators such as myostatin and activins have been found to suppress Akt/mTOR activation through Smad2/3[47,48].

**Fig. 3 TAK1 stimulates translational machinery and augments protein synthesis in cultured myotubes.** Primary myotubes were transduced with either AAV6-GFP or a combination of AAV6-TAK1 and AAV6-TAB1 vectors for 48 h. **a** Representative images of GFP and TAK1 and TAB1 overexpressing myotube cultures after immunostaining for myosin heavy chain (MyHC) using MF20 antibody. Nuclei were counterstained with DAPI. Scale bar, 50 μm. **b** Representative immunoblots and **c** densitometry analysis showing levels of p-rpS6, rpS6, p-eIF4E, eIF4E, p-Mnk1, Mnk1, p-ERK1/2, ERK1/2, p-p90RSK, p90RSK, p-mTOR, mTOR, p-Smad1/5/8, Smad1, TAK1, and TAB1 in cultures transduced with AAV6-GFP alone or a combination of AAV6-TAB1 and AAV6-TAK1. GAPDH was used as a normalizing control. **d** In the same experiment, the GFP and TAK1 with TAB1 overexpressing myotubes were incubated with 100 μM puromycin for 30 min. The myotubes were then collected and protein extracts made were subjected to western blot using puromycin antibody. Immunoblots showing levels of puromycin-tagged proteins and an unrelated protein, GAPDH. **e** Densitometry analysis of levels of puromycin-tagged protein in GFP and TAK1 with TAB1 overexpressing myotube cultures. $N = 3$ in each group. Data presented as mean ± SEM. *$p < 0.05$, values significantly different from corresponding cultures transduced with AAV6-GFP by unpaired t-test. **f** Cultured primary myotubes transduced with AAV6-GFP or AAV6-TAK1 and AAV6-TAB1 were treated with vehicle alone or 0.5 μM rapamycin for 24 h. The cultures were then treated with 100 μm of puromycin for 30 min. Immunoblots presented here demonstrate the levels of p-rpS6, rpS6, p-TAK1 and puromycin-tagged proteins. **g** Densitometry analysis of puromycin-tagged proteins in GFP and TAK1- with TAB1-overexpressing myotube cultures treated with vehicle alone or rapamycin. $N = 3$ in each group. Data presented as mean ± SEM. *$p < 0.05$, values significantly different from corresponding control cultures treated with vehicle analyzed by two-way ANOVA followed by Tukey's multiple comparison test. #$p < 0.05$, values significantly different from cultures transduced with AAV6-GFP and treated with rapamycin by analyzed by two-way ANOVA followed by Tukey's multiple comparison test.

Conversely, BMP family ligands integrate Smad1 and Smad4 to positively regulate skeletal muscle mass, particularly in denervated muscle[11,49]. A few studies have suggested that TAK1 has a role in the regulation of Smad signaling during osteogenesis, chondrogenesis, and cancer growth[21,50–55]. Whether such an interplay between TAK1 and Smad signaling exists in the regulation of skeletal muscle mass has never been investigated. Our results showed a significant increase in the phosphorylation of both BMP specific Smad1/5/8 and TGF-β specific Smad2 in Tak1mKO mice compared to littermate Tak1fl/fl mice (Fig. 5a, b). We also observed a significant increase in the levels of total Smad1 and Smad2 in skeletal muscle of Tak1mKO mice compared to Tak1fl/fl mice. However, there was no noticeable difference in the levels of Smad4 and Smad6 (Fig. 5a, b). To further understand the mechanism of activation of R-smads (i.e. Smad1/5/8 and Smad2/3) after TAK1 inactivation in skeletal muscle of mice, we measured mRNA levels of BMP and TGF-β subfamily ligands and their receptors. Interestingly, ALK3 (*Bmpr1a*), ALK6 (*Bmpr1b*) and *Bmpr2*, the receptors involved in the activation of BMP-Smad1/5/8 axis as well as the receptors ALK4 (*Acvr1b*), ALK7 (*Acvr1c*), and *Tgfbr2* associated with activating the TGFβ-Smad2/3 arm were found to be significantly upregulated in GA muscle of Tak1mKO mice compared to Tak1fl/fl mice (Fig. 5c, d). Furthermore, there was a significant increase in the mRNA levels of *Bmp4*, *Bmp7*, *Bmp8a*, *Bmp8b*, *Bmp13* (*Gdf6*), *Bmp14* (*Gdf5*) and *Bmp15* in GA muscle of Tak1mKO mice compared to Tak1fl/fl mice (Fig. 5e). Additionally, the mRNA levels of *Tgfb1*, *Tgfb2* and *Tgfb3*, the ligands for activating Smad2 axis, were also found to be significantly increased in the Tak1mKO mice compared to Tak1fl/fl mice (Fig. 5f). Intriguingly, we also observed a significant increase in mRNA levels of Follistatin-288 (*Fst*), a known inducer of skeletal muscle hypertrophy[48], in GA muscle of Tak1mKO mice compared to Tak1fl/fl mice (Fig. 5g). By contrast, mRNA levels of *Bmp2*, *Bmp3*, *Bmp5*, *Bmp6*, *Bmp11* (*Gdf11*), *Bmp12* (*Gdf7*), Myostatin (*Mstn*), Inhibin beta A (*Inhba*), *Acvr2a* and *Acvr2b* receptors remained comparable in GA muscle of Tak1fl/fl and Tak1mKO mice (Supplementary Fig. 6).

**TAK1 interacts with Smad1 upon denervation to prevent excessive muscle atrophy.** TAK1-mediated Smad signaling is known to be cell type specific and context-dependent[21,50–55]. In skeletal muscle, activation of Smad 1/5/8 signaling axis acts as a compensatory pathway to prevent excessive muscle wasting during denervation[1,14]. Previously, we observed that targeted deletion of TAK1 exacerbates denervation-induced muscle atrophy in mice[23]. Based on our results in this study showing a drastic imbalance in Smad signaling, activation of HDAC4-myogenin

axis, and disruption of NMJs upon Tak1 deletion, we speculated that TAK1 signaling regulates skeletal muscle mass in response to denervation. To test this hypothesis, we first performed a co-immunoprecipitation experiment to evaluate interaction between TAK1 and Smad1 protein in denervated skeletal muscle of mice. Results showed a significant enrichment of TAK1 protein after immunoprecipitation with Smad1 antibody (Fig. 6a, b) and enrichment of Smad1 when immunoprecipitation was performed with TAK1 antibody (Fig. 6c). By contrast, we did not find any interaction between TAK1 and Smad4 in control or denervated muscle (Supplementary Fig. 7a).

Published reports suggest that interaction between TAK1 and Smad proteins fine tunes the outcome of Smad signaling[50,51]. In denervated muscle, BMP ligands bind to their receptors and induce the phosphorylation of Smad1 which suppresses gene expression of various atrogenes[1]. We hypothesize that TAK1 and Smad1 interaction is necessary for the Smad1-mediated prevention of excessive muscle atrophy during denervation. To test this hypothesis, littermate Tak1fl/fl and Tak1mKO mice were treated with tamoxifen for four consecutive days to delete Tak1 gene in skeletal muscle and the mice were then subjected to denervation surgery. At day 7 post-denervation, we found that the denervation-induced loss in TA and GA muscle weight (normalized by body weight) was higher in Tak1mKO mice compared with Tak1fl/fl mice (Supplementary Fig. 7b). Western blot analysis showed that the levels of MAFbx and MuRF1 were significantly higher in denervated muscle of Tak1mKO mice compared to corresponding denervated muscle of Tak1fl/fl mice (Fig. 6d, e). Moreover, there was a significant increase in the relative mRNA level of *Musa1* (*Fbxo30*) in denervated GA muscle of Tak1mKO mice compared with denervated muscle of Tak1fl/fl mice (Fig. 6f). In addition, there was also a significant increase in the amount of ubiquitin-conjugated proteins in the denervated GA muscle of Tak1mKO mice compared to Tak1fl/fl mice (Fig. 6d, g). Interestingly, we also observed a significant increase in p-Smad1/5/8 levels in the denervated muscle from Tak1mKO mice compared to denervated muscle of Tak1fl/fl mice. By contrast, there was no significant difference in the levels of Smad1, p-Smad2, Smad2, Smad4 and Smad6 between denervated muscle of Tak1mKO and Tak1fl/fl mice (Fig. 6d, h). BMP-mediated phosphorylation of Smad1 limits MAFbx, MuRF1 and MUSA1 expression during denervation[1,14]. Our results suggest that TAK1 facilitates Smad1-mediated suppression of MAFbx, MuRF1, and MUSA1 expression.

We also evaluated the activation of key regulators of protein translation machinery. A significant increase in levels of p-eIF4E and p-rpS6 protein was observed in denervated GA muscle compared to

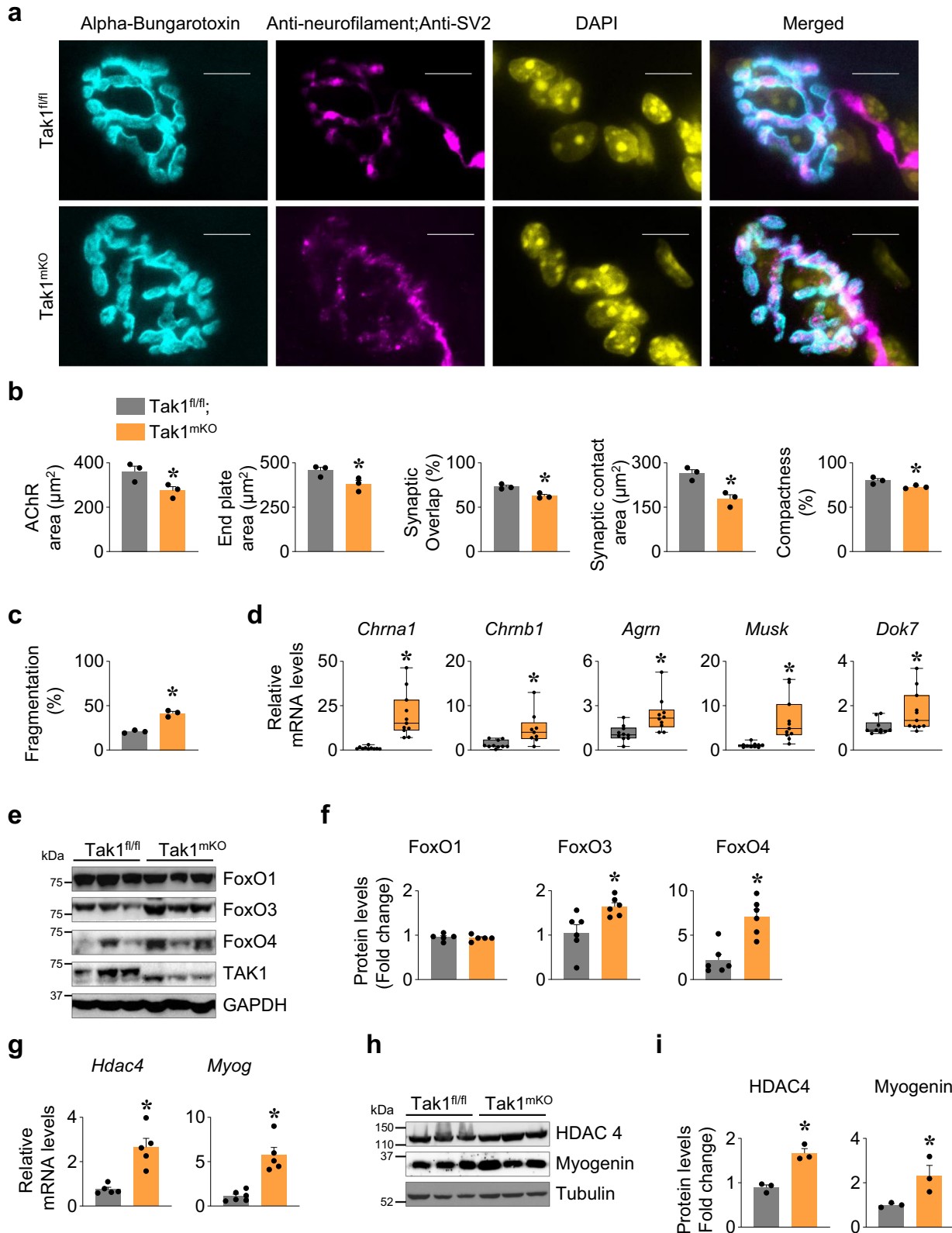

innervated GA muscles of Tak1$^{fl/fl}$ mice (Fig. 6d, h). Interestingly, inactivation of TAK1 significantly reduced the denervation-induced increase in p-eIF4E levels but not p-rpS6 in GA muscle of mice (Fig. 6d, h). Altogether, these results demonstrate that TAK1 interacts with Smad1 to inhibit gene expression of muscle-specific E3 ubiquitin ligases and stimulates the protein translational machinery in denervated skeletal muscle of mice.

**TAK1 regulates subcellular distribution of Smad proteins in denervated skeletal muscle.** In addition to the phosphorylation status of R-Smads (i.e. Smad1/5/8 and Smad2/3), their ability to form multi-protein complexes with Co-Smad (i.e. Smad 4) and I-Smads (i.e. Smad 6 and 7) as well as the subcellular locations of Smad complex influence and determine the ultimate consequence of Smad signaling[56,57]. Therefore, we studied the subcellular

**Fig. 4 Targeted inactivation of Tak1 leads to NMJ derangement.** Littermate Tak1$^{fl/fl}$ and Tak1$^{mKO}$ mice were administered with tamoxifen i.p. (75 mg/kg bodyweight) for 4 days and then fed with tamoxifen containing diet for 24 days to delete Tak1 gene. 30 μm longitudinal sections of TA muscle were generated and stained with α-Bungarotoxin-Alexa 488, anti-neurofilament (clone 2H3) and anti-synaptic vesicle 2 A protein (anti-SV2), and DAPI. **a** Representative immunofluorescence images of NMJ in Tak1$^{fl/fl}$ and Tak1$^{mKO}$ mice. Scale bar, 10 μm. Morphometric analysis using ImageJ software showing **b** AChR area, End plate area, percent synaptic overlap, synaptic contact area and percent compactness. **c** Quantification of fragmentation at the post synaptic end plate in Tak1$^{fl/fl}$ and Tak1$^{mKO}$ mice. $n = 3$ mice in each group. **d** Relative mRNA levels of NMJ components: AChRα (*Chrna1*), AChRβ (*Chrnb1*), Agrin (*Agrn*), *Musk* and *Dok7* in GA muscle of Tak1$^{fl/fl}$ and Tak1$^{mKO}$ mice measured by performing qPCR. $n = 9$–11 mice in each group. **e** Representative immunoblots, and **f** densitometry analysis demonstrating levels of FoxO1, FoxO3, FoxO4, TAK1, and unrelated protein GAPDH in GA muscle of Tak1$^{fl/fl}$ and Tak1$^{mKO}$ mice. $n = 5$–6 mice in each group. **g** Relative mRNA levels of *Hdac4* and *Myog* in GA muscle of Tak1$^{fl/fl}$ and Tak1$^{mKO}$ mice assayed by performing qPCR. $n = 4$–6 mice in each group. **h** Representative immunoblots and **i** densitometry analysis showing levels of HDAC4, myogenin and an unrelated protein α-Tubulin in GA muscle of Tak1$^{fl/fl}$ and Tak1$^{mKO}$ mice. $n = 3$ mice in each group. Data presented as mean ± SEM. *$p < 0.05$, values significantly different from corresponding Tak1$^{fl/fl}$ mice by unpaired *t*-test.

distribution of Smads in skeletal muscle of Tak1$^{fl/fl}$ and Tak1$^{mKO}$ mice in response to denervation. We measured levels of various Smad proteins in the nuclear and cytosolic fractions of innervated and denervated GA muscle of Tak1$^{fl/fl}$ and Tak1$^{mKO}$ mice. Results showed that there was a considerable increase in the levels of p-Smad1/5/8 in nuclear fraction of denervated GA muscle of both Tak1$^{fl/fl}$ and Tak1$^{mKO}$ mice compared to contralateral control GA muscle. (Fig. 7a, c). Interestingly, we also found that there was a significant increase in nuclear Smad4 levels in denervated GA muscle of Tak1$^{fl/fl}$ mice compared to innervated GA muscle of Tak1$^{fl/fl}$ mice (Fig. 7a, d). However, no such increase in the Smad4 levels in nuclear fraction of denervated GA muscle of Tak1$^{mKO}$ mice was noticeable (Fig. 7a, d).

Significantly higher levels of Smad6 were observed in the nuclear fraction of denervated GA muscle of Tak1$^{mKO}$ mice compared with denervated GA muscle of Tak1$^{fl/fl}$ mice (Fig. 7a, e). This difference in nuclear Smad6 between Tak1$^{fl/fl}$ and Tak1$^{mKO}$ was also observed in innervated muscle (Fig. 7a, e). In the cytoplasmic fraction, p-Smad1/5/8 was significantly elevated in denervated GA muscle of Tak1$^{mKO}$ mice compared to control GA muscle (Fig. 7b, c). There was no noticeable difference in Smad4 levels in the cytosolic fractions of denervated GA muscle of Tak1$^{fl/fl}$ and Tak1$^{mKO}$ mice (Fig. 7b, d). Correspondingly, the cytosolic levels of Smad6 were found to be significantly lower in the denervated GA muscle of Tak1$^{mKO}$ mice compared to all other genotypes (Fig. 7b, e). GAPDH protein is present in both cytoplasm and nuclear extracts whereas Lamin B1 is used as a loading control for nuclear extracts. Our results showed that GAPDH was enriched in cytoplasmic extracts whereas Lamin B1 was predominantly enriched in nuclear extracts confirming the purity of cytoplasmic and nuclear extracts (Fig. 7a, b). Our results suggest that TAK1 regulates the spatial distribution of Smad1, Smad4 and Smad6 in denervated muscle. However, further investigations are needed to understand how TAK1 supports the nuclear translocation of Smad4 and cytosolic retention of Smad6 protein in denervated muscle.

**BMP ligands induce the phosphorylation of TAK1 in skeletal muscle.** We first measured how the levels of some key BMP ligands were affected in skeletal muscle of Tak1$^{fl/fl}$ and Tak1$^{mKO}$ mice in response to denervation. Our qPCR analysis showed that the mRNA levels of both *Bmp13* and *Bmp14* were drastically increased in skeletal muscle during denervation. Moreover, the levels of *Bmp13* were significantly higher in the denervated muscle of Tak1$^{mKO}$ mice compared to corresponding denervated muscle of Tak1$^{fl/fl}$ mice (Fig. 8a, b). Using cultured myotubes, we next sought to investigate whether BMP ligands activate TAK1 and whether TAK1 is required for the BMP-induced phosphorylation of Smad1/5/8 in skeletal muscle. Mouse primary myoblasts were differentiated into myotubes by incubation in differentiation medium for 72 h. The myotubes were then pretreated with 5Z-7-Oxozeanol

(5-Z7O), a well-known pharmacological inhibitor of enzymatic activity of TAK1[58,59] followed by addition of recombinant BMP7 or BMP13 protein. Western blot analysis showed that both treatment with either BMP7 or BMP13 increased the phosphorylation of TAK1 and Smad1/5/8 in cultured myotubes (Fig. 8c) However, the BMP7- or BMP13-induced activation of p-Smad1/5/8 was considerably reduced in myotubes pretreated with 5-Z7O. As expected, 5-Z7O blunted the phosphorylation of TAK1 in response to BMP7 or BMP13 (Fig. 8c).

As a complimentary approach, we also investigated the effect of knockdown of TAK1 in the BMP-induced activation of Smad1/5/8 in myotubes. Mouse primary myotubes were transduced with adenoviral constructs expressing either a scrambled short hairpin RNA (shRNA) or *Tak1* shRNA along with GFP (Fig. 8d). After 48 h, the mRNA levels of *Tak1* were found to be drastically reduced in cultures transduced with adenoviral constructs expressing *Tak1* shRNA (Fig. 8e). In a parallel experiment, the myotubes expressing control or *Tak1* shRNA were treated with recombinant BMP7 or BMP13 protein for 30 min and the levels of p-Smad1/5/8 were measured by performing western blot. Results showed that knockdown of TAK1 reduced the BMP7- or BMP13-induced phosphorylation of Smad1/5/8 in cultured myotubes. Our western blot analysis also confirmed that TAK1 protein was considerably reduced in cultures transduced with Ad.TAK1 shRNA (Fig. 8f). Taken together, these results suggest that TAK1 contributes to the BMP-induced Smad1 signaling in skeletal muscle.

**Forced activation of TAK1 attenuates denervation-induced muscle wasting.** Our preceding results identified a critical regulatory role of TAK1 in preventing excessive loss of muscle mass during denervation. To further validate the growth promoting effects of TAK1 and its role in preventing excessive muscle wasting during neurogenic atrophy, we studied the effect of forced activation of TAK1 in denervation-induced muscle atrophy. Wild-type mice were given intramuscular injection of either AAV6-GFP ($2.5 \times 10^{10}$ vg) or a combination of AAV6-TAK1 ($1.25 \times 10^{10}$ vg) and AAV6-TAB1 ($1.25 \times 10^{10}$ vg) in both TA muscle. After 14 days of intramuscular injection of AAVs, denervation surgery was performed on left leg whereas right leg was sham-operated. After 7d or 14d, the mice were euthanized, and TA muscle was isolated and used for morphometric and biochemical analyses. Interestingly, overexpression of TAK1 and TAB1, but not GFP, attenuated the loss in wet weight of TA muscle at 7 day post-denervation (Fig. 9a, b). We next performed immunostaining for dystrophin protein followed by quantification of myofiber CSA (Fig. 9c). This analysis showed that average myofiber CSA was significantly higher in AAV6-TAK1 and AAV6-TAB1 injected 7d-denervated TA muscle compared to AAV6-GFP injected 7d-denervated TA muscle of mice (Fig. 9d). Indeed, the average myofiber CSA of denervated TA muscle

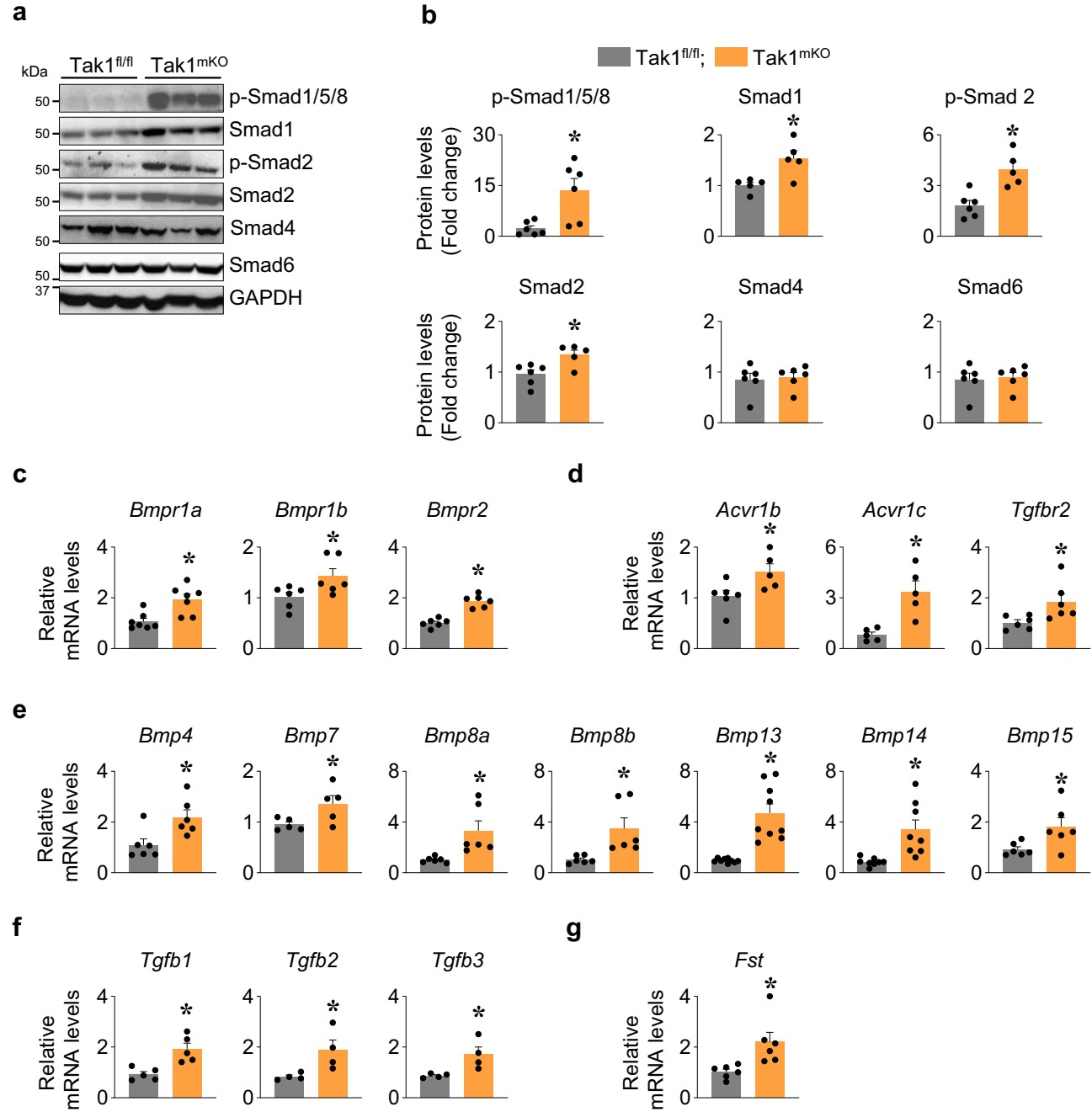

**Fig. 5 Targeted inactivation of TAK1 disrupts Smad signaling in skeletal muscle.** Littermate Tak1^fl/fl and Tak1^mKO mice were administered with tamoxifen (75 mg/kg bodyweight) for 4 days and then fed with tamoxifen containing diet for 24 days to delete Tak1 gene. Finally, GA muscle of the mice was harvested and used for analysis by performing western blotting and qPCR analysis. **a** Representative immunoblots, **b** densitometry analysis showing levels of p-Smad1/5/8, Smad1, p-Smad2, Smad2, Smad4, and Smad6 in GA muscle of Tak1^fl/fl and Tak1^mKO mice. n = 5–6 in each group. Relative mRNA levels of **c** BMP specific receptors ALK3 (*Bmpr1a*), ALK6 (*Bmpr1b*), *Bmpr2*, **d** TGFβ specific receptors ALK4 (*Acvr1b*), ALK7 (*Acvr1c*), *Tgfbr2*, **e** BMP subfamily ligands, *Bmp4, Bmp7, Bmp8a, Bmp8b, Bmp13 (Gdf6), Bmp14 (Gdf5), Bmp15*, **f** TGFβ subfamily ligands, *Tgfb1, Tgfb2, Tgfb3*, and **g** Follistatin-288 (*Fst*) in GA muscle of Tak1^fl/fl and Tak1^mKO mice assayed by qPCR. n = 4–9 per group. Data presented here are mean ± SEM. *p < 0.05, values significantly different from corresponding Tak1^fl/fl mice by unpaired *t*-test.

injected with AAV6-TAK1 and AAV6-TAB1 was comparable to innervated AAV6-GFP injected TA and also to contralateral sham-operated AAV6-TAK1 and AAV6-TAB1 co-injected TA (Fig. 9d). Relative frequency distribution of myofibers also demonstrated higher percentage of myofibers with larger CSA in AAV6-TAK1 and AAV6-TAB1 injected TA muscle compared to AAV6-GFP injected TA muscle at day 7 following denervation

(Fig. 9e, f). In another experiment, we observed that average myofiber CSA was also significantly higher in TAK1- and TAB1-overexpressing TA muscle compared with GFP overexpressing TA muscle at day 14 post-denervation (Supplementary Fig. 8a–d). H&E staining of TA muscle sections showed that overexpression of TAK1 and TAB1 did not produce any overt effect in control or 14d-denervated muscle of mice (Supplementary Fig. 8e).

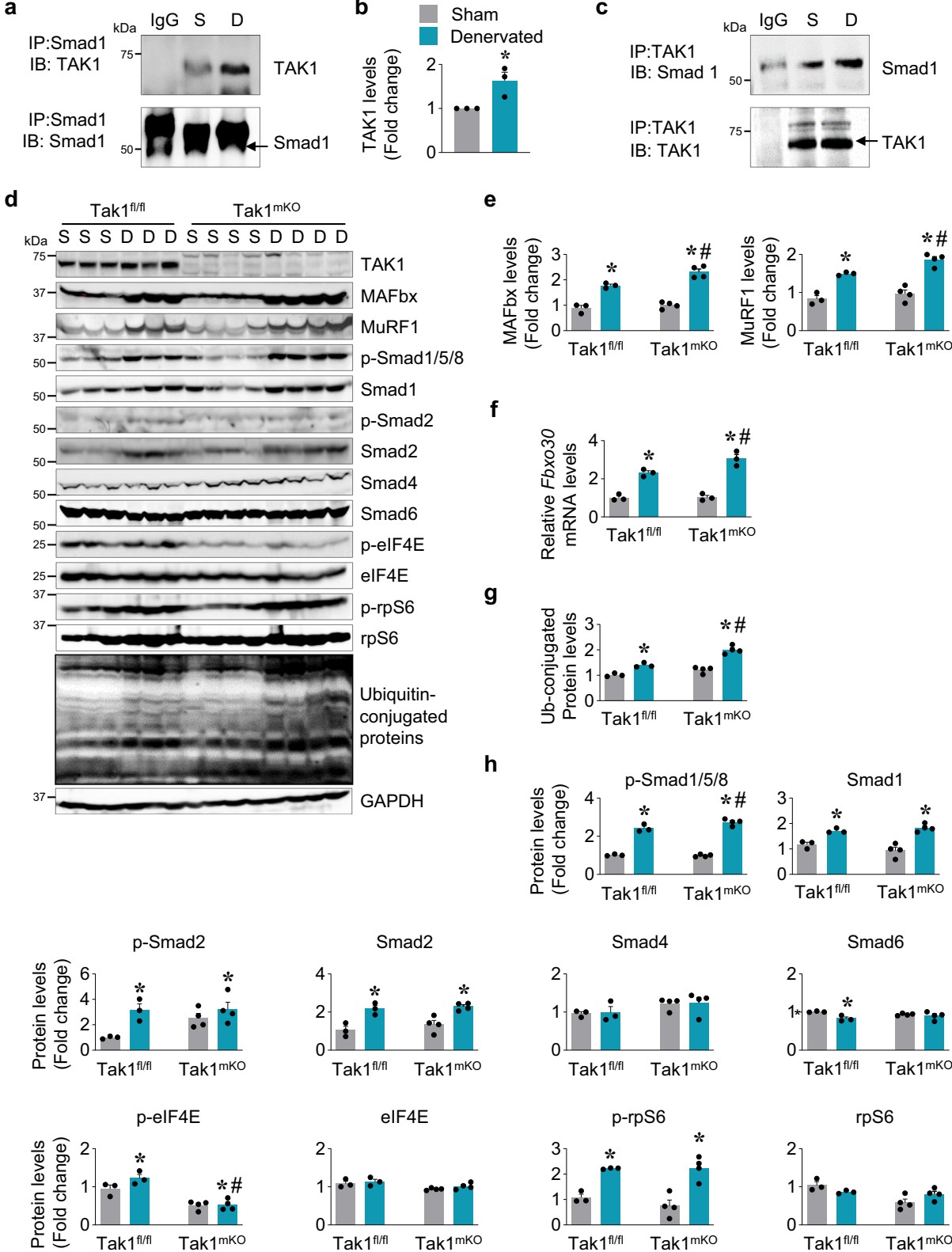

Western blot analysis showed similar levels of p-mTOR, p-Smad1/5/8, p-Smad2, p-rpS6 and total Smad1, total Smad2, and total rpS6 between innervated and denervated TA muscle receiving either AAV6-GFP or AAV6-TAK1 and AAV6-TAB1 (Fig. 9g, h). Interestingly, there were significant increase in the levels of p-Mnk1 and p-eIF4E in denervated TA muscle injected with AAV6-TAK1 and AAV6-TAB1 compared to denervated muscle injected with AAV6-GFP (Fig. 9g, h). Taken together, these results suggest that forced activation of TAK1 mitigates neurogenic muscle atrophy potentially through activation of Mnk1-dependent stimulation of protein translational machinery.

**Fig. 6 TAK1 regulates Smad signaling in denervated muscle to alleviate atrophy.** C57BL/6 mice were subjected to sham or denervation surgery for 7d followed by isolation of hind limb muscle and biochemical analysis. **a** Immunoblots after immunoprecipitation of GA muscle extracts with Smad1 antibody followed by immunoblotting with TAK1 or Smad1 antibody. **b** Fold increase in TAK1 enrichment in anti-Smad1 immunoprecipitated complex. $n = 3$ in each group. $*p < 0.05$, values significantly different from sham-operated muscle by unpaired $t$-test. **c** Immunoprecipitation of GA muscle extracts with TAK1 antibody followed by immunoblotting with Smad1 or TAK1 antibody. Representative immunoblots are presented here. **d** Immunoblots showing protein levels of TAK1, MAFbx, MuRF1, p-Smad1/5/8, Smad1, p-Smad2, Smad2, Smad4, Smad6, p-eIF4E, eIF4E, p-rpS6, rpS6, ubiquitin-conjugated proteins, and unrelated protein GAPDH in control and denervated GA muscle of Tak1[fl/fl] and Tak1[mKO] mice. **e** Densitometry analysis showing protein levels of MAFbx and MuRF1. **f** Relative mRNA level of *Musa1* (*Fbxo30*) assayed by performing qPCR. $n = 3$–4 mice in each group. **g** Fold change in the levels of ubiquitin-conjugated proteins in sham-operated and denervated muscle of Tak1[fl/fl] and Tak1[mKO] mice. **h** Densitometry analysis of immunoblots (from d part) to measure relative levels of p-Smad1/5/8, Smad1, p-Smad2, Smad2, Smad4, Smad6, p-eIF4E, eIF4E, p-rpS6, and rpS6 in sham and denervated GA muscle of Tak1[fl/fl] and Tak1[mKO] mice. $n = 3$–4 mice per group. Data are presented as mean ± SEM and was analyzed by two-way ANOVA followed by Tukey's multiple comparison test. $*p < 0.05$, values significantly different from corresponding sham-operated muscle. $\#p < 0.05$, values significantly different from denervated muscle of Tak1[fl/fl] mice. S, Sham; D, Denervated.

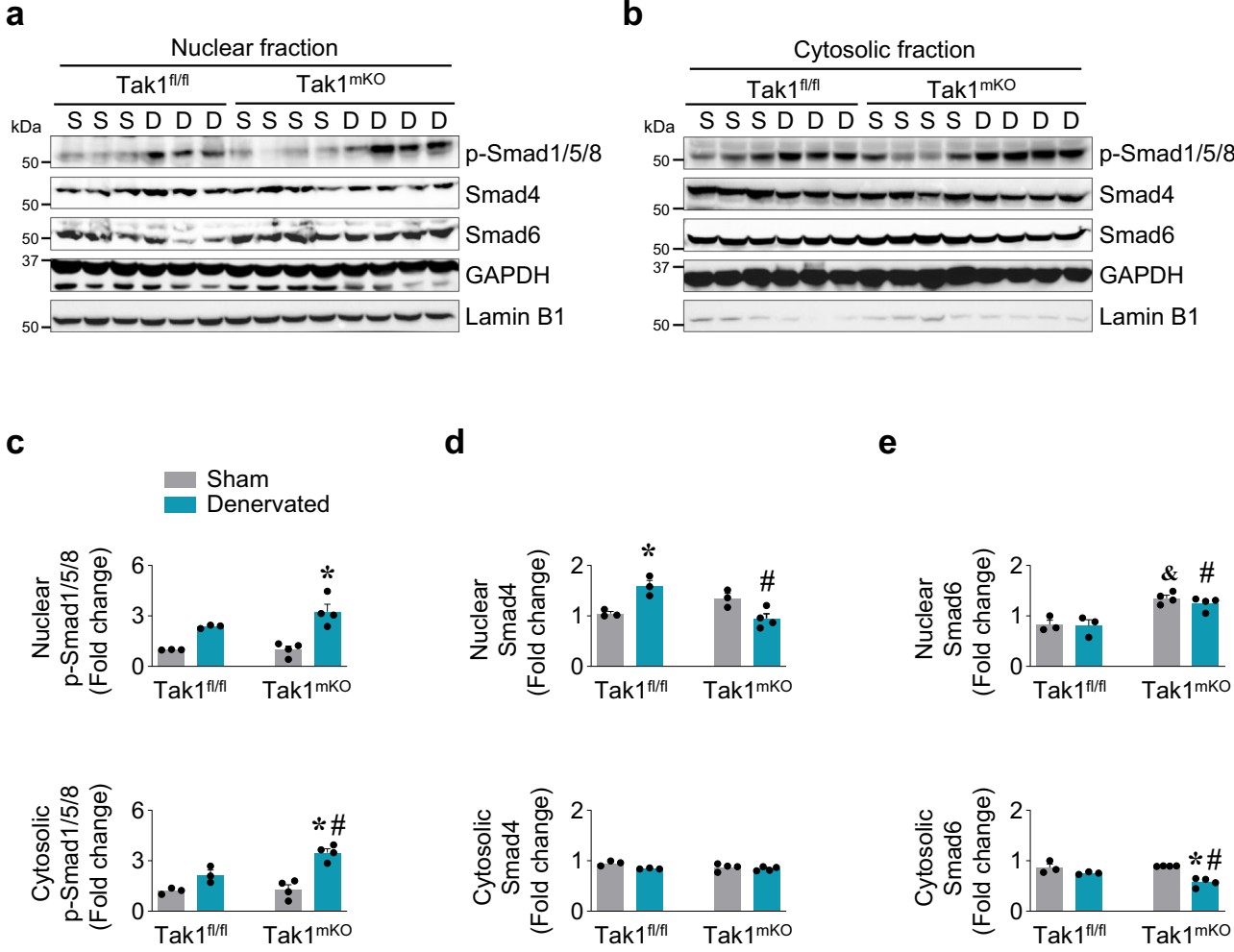

**Fig. 7 TAK1 regulates subcellular distribution of Smad proteins in denervated muscle.** Sham-operated and 7d-denervated GA muscle isolated from Tak1[fl/fl] and Tak1[mKO] mice was processed to prepare nuclear and cytoplasmic extracts. Immunoblots presented here demonstrate **a** nuclear, and **b** cytosolic levels of p-Smad1/5/8, Smad4 and Smad6 in GA muscle of Tak1[fl/fl] and Tak1[mKO] mice. Lamin B1 and GAPDH were used as loading controls for nuclear and cytoplasmic fractions, respectively. Densitometry analysis showing fold changes in nuclear and cytosolic protein levels of **c** p-Smad1/5/8, **d** Smad4, and **e** Smad6 in sham and denervated GA muscle of Tak1[fl/fl] and Tak1[mKO] mice. $n = 3$–4 per group. Data are presented as mean ± SEM and was analyzed by two-way ANOVA followed by Tukey's multiple comparison test. $*p < 0.05$, values significantly different from sham-operated corresponding control muscle. $\#p < 0.05$, values significantly different from denervated muscle of Tak1[fl/fl] mice. $\&p < 0.05$ values significantly different from sham-operated Tak1[fl/fl] mice. S, Sham; D, Denervated.

## Discussion

Skeletal muscle wasting occurs in several disease states and neuromuscular disorders[1,2,38]. It is now increasingly clear that a complex signaling network regulates skeletal muscle mass in diverse conditions. TAK1 along with its binding partners TAB1, TAB2, and TAB3 constitute an important signalosome, which fine-tunes the activation of several intracellular signaling pathways during embryonic development and throughout the lifespan

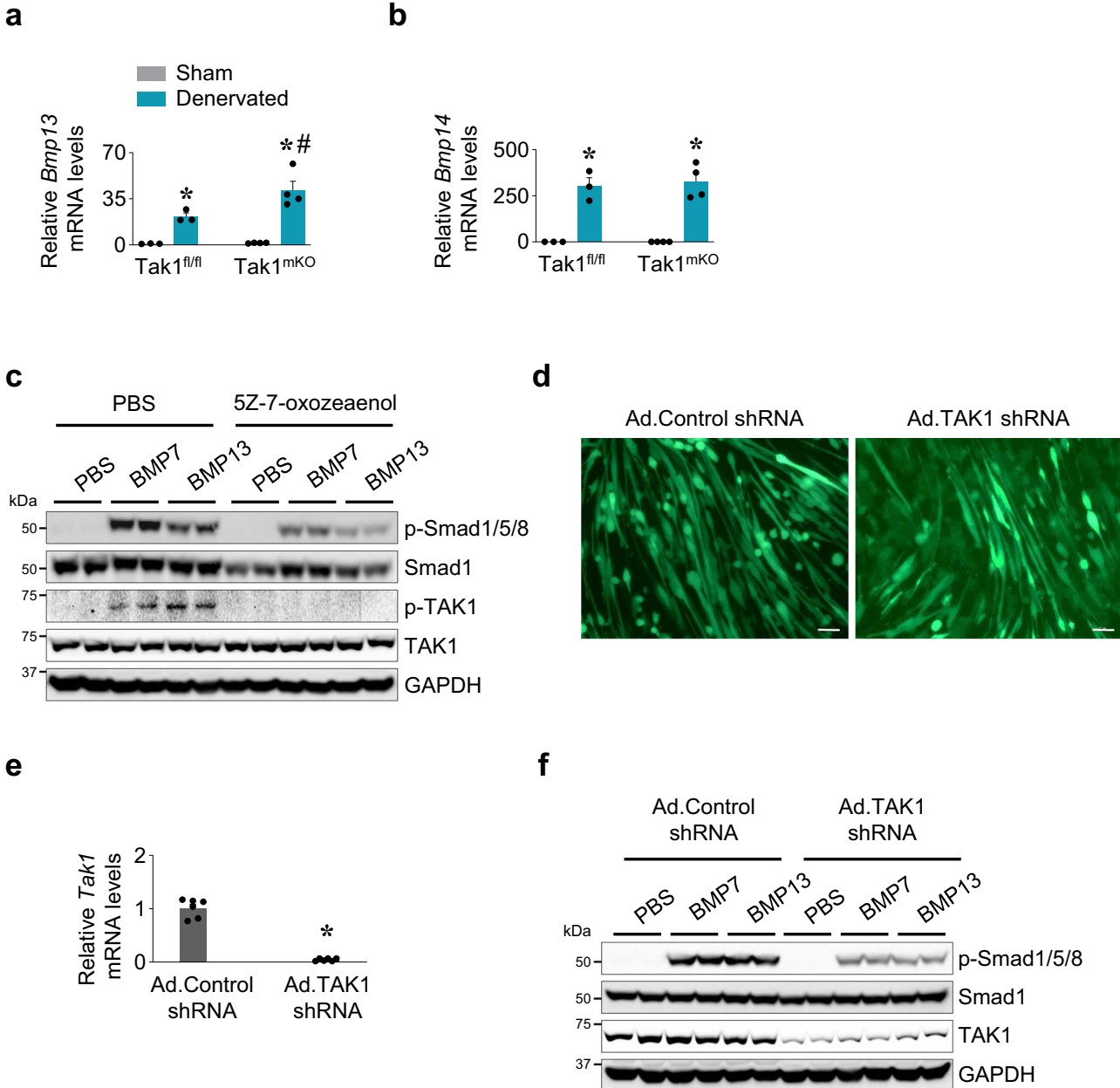

**Fig. 8 BMP protein phosphorylates TAK1 in skeletal muscle.** Relative mRNA levels of **a** BMP13 and **b** BMP14 in sham and 7d-denervated GA muscle of Tak1fl/fl and Tak1mKO mice. $n = 3–4$ per group. Data are presented as mean ± SEM and was analyzed by two-way ANOVA followed by Tukey's multiple comparison test. *$p < 0.05$, values significantly different from sham-operated corresponding control muscle. #$p < 0.05$, values significantly different from denervated muscle of Tak1fl/fl mice. **c** Primary myoblasts isolated from wild-type mice were differentiated into myotubes by incubation in differentiation medium. The myotubes were treated with 2 μM 5Z-7oxozeaenol or vehicle alone for 2 h followed by treatment with recombinant mouse BMP7 (100 ng/ml), recombinant mouse BMP13 (150 ng/ml), or vehicle (PBS) for 30 min. Representative immunoblots showing levels of p-Smad1/5/8, Smad1, p-TAK1, and total TAK1. GAPDH was used as loading control. **d** Primary myotubes were transduced with adenoviral vector expressing scrambled shRNA (Ad.Control shRNA) or TAK1 shRNA (Ad.TAK1 shRNA) for 48 h. The myotubes were then treated with recombinant mouse BMP7 (100 ng/ml), BMP13 (150 ng/ml) or vehicle (PBS) for 30 min. Representative photomicrographs presented here confirm transduction of myotubes by adenoviral vectors. Scale bar, 50 μm. **e** Relative mRNA levels of TAK1 in myotube cultures transduced with Ad.Control shRNA and Ad.TAK1 shRNA vectors. n = 6 in each group. Data are presented as mean ± SEM. *$p < 0.05$, values significantly different from corresponding control shRNA-expressing cultures by unpaired t-test. **f** Primary myotubes transduced with Ad.Control shRNA or Ad.TAK1 shRNA vector were treated with recombinant mouse BMP7 (100 ng/ml), BMP13 (150 ng/ml), or vehicle (PBS) for 30 min followed by performing western blot. Representative immunoblots presented here demonstrate the levels of p-Smad1/5/8, Smad1, TAK1, and an unrelated protein, GAPDH in control and TAK1 knocked down cultures treated with BMP7 or BMP13.

of an organism. In contrast, aberrant activation of TAK1 can lead to various pathological conditions including cancer and tissue degeneration[18,20]. TAK1 is a known upstream kinase that regulates the activation of catabolic signaling pathways such as NF-κB and p38 MAPK in response to inflammatory cytokines and bacterial products however, we surprisingly discovered that TAK1 is essential for the maintenance of skeletal muscle mass and overload-induced skeletal muscle growth[23]. In this study, we report that supraphysiological activation of TAK1 is sufficient to induce myofiber hypertrophy. More importantly, we found that

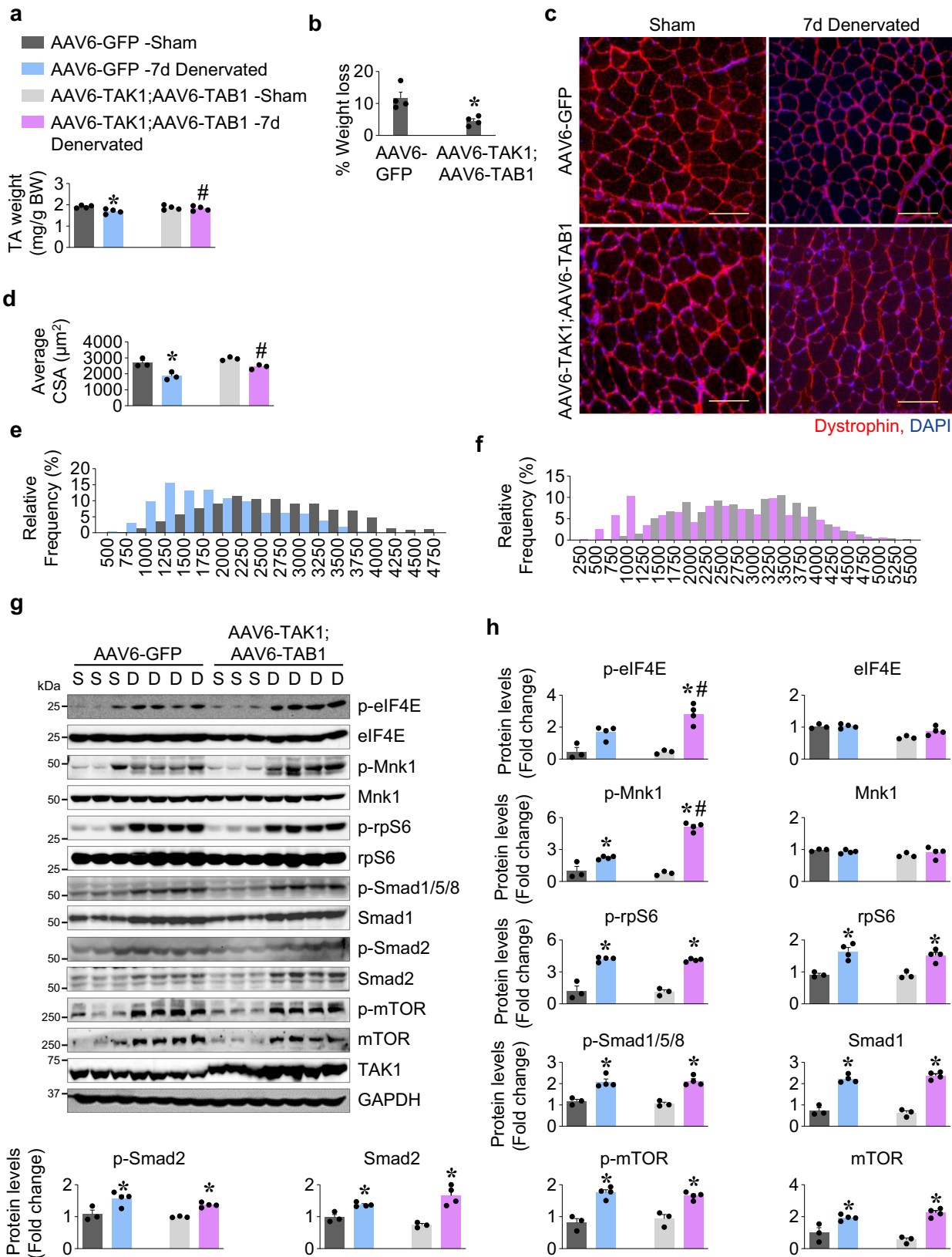

TAK1 is essential for maintenance of NMJs and forced activation of TAK1 mitigates denervation-induced skeletal muscle atrophy in adult mice.

Muscle growth is sustained by increased protein synthesis and organelle biogenesis. In eukaryotes, the rate of protein synthesis is dependent on the heterotrimeric eIF4F complex formation at the translation initiation stage[60,61]. Availability of eIF4E is a critical step for the eIF4F complex formation that directly binds the 5'Cap of the mRNA with other associated factors and initiate translation[62,63]. The activity of eIF4E-mediated protein synthesis is tightly controlled, and it remains inactive when bound to eIF4E-binding proteins (4E-BPs). In response to growth stimuli,

**Fig. 9 Forced activation of TAK1 mitigates denervation-induced muscle atrophy.** Both side TA muscle of adult wild-type mice were given intramuscular injection of AAV6-GFP ($2.5 \times 10^{10}$ vg) or a combination of AAV6-TAK1 ($1.25 \times 10^{10}$ vg) and AAV6-TAB1 ($1.25 \times 10^{10}$ vg) particles. After 14 days, the left hind limb muscles of the mice were denervated by transecting the sciatic nerve whereas right side was sham operated. On day 7 post denervation, the mice were euthanized and TA muscles were harvested and analyzed. **a** Wet weight of TA muscle normalized with body weight (BW), and **b** percentage loss of wet weight of AAV6-GFP or AAV6-TAK1 and AAV6-TAB1 injected TA muscle after 7d of denervation. Data are presented as mean ± SEM and was analyzed by unpaired *t*-test. **c** Representative photomicrographs of TA muscle sections after staining with anti-dystrophin and DAPI. Scale bar, 100 μm. **d** Average myofiber cross sectional area (CSA) in sham and 7d-denervated TA muscle injected with AAV6-GFP or AAV6-TAK1 and AAV6-TAB1. **e** Relative frequency distribution of myofiber CSA expressed as percentage in sham and denervated TA muscles injected with AAV6-GFP. **f** Relative frequency distribution of myofiber CSA expressed as percentage in sham and denervated TA muscles injected with AAV6-TAK1 and AAV6-TAB1. **g** Representative immunoblots and **h** densitometry quantification of levels of p-eIF4E, eIF4E, p-Mnk1, Mnk1, p-rpS6, rpS6, p-Smad1/5/8, Smad1, p-Smad2, Smad2, p-mTOR, mTOR, TAK1 and GAPDH protein in sham and denervated TA muscle of mice injected with AAV6-GFP or a combination of AAV6-TAK1 and AAV6-TAB1. *n* = 3–4 mice in each group. Data are presented as mean ± SEM and was analyzed by two-way ANOVA followed by Tukey's multiple comparison test. *$p < 0.05$, values significantly different from sham-operated corresponding control TA muscle. #$p < 0.05$, values significantly different from 7d-denervated TA muscle injected with AAV6-GFP. S, Sham; D, Denervated.

mTORC1-mediated phosphorylation of 4E-BPs frees eIF4E to initiate protein translation. However, the activation of mTORC1 pathway is not the only mechanism to initiate protein synthesis. For example, phosphorylation of eIF4E at Ser209 by Mnk1 and Mnk2 also stimulates protein synthesis in response to mitogenic stimuli[64,65]. Additionally, ribosomal protein S6 (rpS6) is a dominant determinant of ribosomal biogenesis and protein synthesis. Extracellular growth stimuli trigger rapid mTOR-p70S6K-mediated phosphorylation of rpS6 to positively influence cellular size, proliferation, or differentiation. In addition, several mitogens and growth factors activate ERK1/2 and p38 MAPKs which directly phosphorylate and activate p90RSK in many cell types. Activated p90RSK phosphorylates rpS6 kinase which, thereafter, initiate Cap-dependent protein translation and ribosomal biogenesis[37]. Our results showed that overexpression of TAK1 and TAB1 causes increased phosphorylation of TAK1 within its activation domain. Moreover, activation of TAK1 induces the phosphorylation of p38MAPK, Mnk1, p90RSK, rpS6, and eIF4E and augments protein levels of several other eIFs in skeletal muscle of mice and cultured myotubes (Figs. 2 and 3). Phosphorylation of eIF2α blocks cap-dependent protein translation and suppresses global protein synthesis[66]. Interestingly, a significant reduction in phosphorylation of eIF2α was noticeable in TAK1/TAB1-overexpressing TA muscle. Together, we provide initial evidence that forced activation of TAK1 stimulates protein translational machinery without having much effect on mTOR or p70S6K. These results suggest that the role of mTOR in mediating protein synthesis and skeletal muscle hypertrophy is dispensable and can be achieved through the activation of TAK1.

In denervated muscle, we observed a significant increase in the phosphorylation levels of eIF4E and rpS6 which might be a mechanism to stimulate protein synthesis and prevent excessive muscle loss. Loss of nerve supply in skeletal muscle has been found to increase the rate of protein synthesis[67]. Interestingly, the phosphorylation of eIF4E upon denervation was absent in Tak1 deleted mice (Fig. 6d, h) and was significantly elevated in TAK1-TAB1 overexpressing muscle (Fig. 9g, h). However, the phosphorylation status of rpS6 in denervated muscle was not influenced with Tak1 deletion or overexpression. These findings indicate that eIF4E-mediated protein synthesis during muscle mass is primarily regulated by TAK1.

NF-κB and p38 MAPK signaling have been found to induce muscle wasting in many conditions especially in chronic disease states[1,2]. Proinflammatory cytokines and microbial products activate NF-κB and p38MAPK through upstream activation of TAK1[68]. Indeed, there are recent reports suggesting that the inhibition of TAK1 prevents muscle wasting in animal models of inflammatory diseases and Duchenne muscular dystrophy (DMD) which involve chronic and severe inflammatory immune

response[69–71]. Since TAK1 is also involved in cytokine-induced activation of multiple inflammatory pathways, inhibition of TAK1 in models of chronic inflammation may attenuate muscle wasting through inhibiting proinflammatory signaling pathways. We activated TAK1 specifically in skeletal muscle of wild-type mice which are devoid of any inflammatory response. We found that forced activation of TAK1 stimulates myofiber growth with no sign of inflammation or fibrosis even though TAK1 increases the phosphorylation of p65 subunit of NF-κB and p38MAPK in skeletal muscle (Figs. 1 and 2 and Supplementary Fig. 1e). These results are consistent with published reports suggesting that NF-κB and p38MAPK promote skeletal muscle growth and homeostasis in physiological conditions[69,72]. Therefore, it is possible that the activation of TAK1 alone and its downstream effectors, such as NF-κB and p38MAPK, promote skeletal muscle growth in naive conditions. We observed that TAK1 is activated along with other positive regulators of muscle mass during overload-induced hypertrophy (Fig. 1a, b). Moreover, TAK1 interacts Smad1 in denervated muscle which may be a mechanism to prevent excessive muscle wasting (Fig. 6). Finally, our results demonstrating that the BMP-induced Smad1/5/8 phosphorylation was reduced upon pharmacological inhibition or knockdown of TAK1 in cultured myotubes (Fig. 8) further suggest a growth promoting role of TAK1 in skeletal muscle during catabolic periods.

Sarcopenia and muscle wasting influencing NMJ homeostasis is an enigmatic concept[73]. Our results demonstrating disruption of NMJ morphology and enhanced NMJ gene expression in skeletal muscle of Tak1[mKO] indicate that TAK1 plays an important role in maintaining NMJ health (Fig. 4). Recent studies have indicated that autophagy and ubiquitin-proteasome system regulate the turnover of nicotinic acetylcholine receptors in skeletal muscle during catabolic period[73–75]. FoxO family of transcription factors mediate denervation-induced muscle wasting through augmenting the gene expression of components of autophagy-lysosomal system and E3 ubiquitin ligases, such as MAFbx and MuRF1[76]. Furthermore, HDAC4-Dach2-myogenin axis augments the levels of MAFbx and MuRF1 during neurogenic atrophy[1,2]. Our results demonstrate that targeted inactivation of TAK1 increases the levels of FoxO proteins, HDAC4, and myogenin which are the hallmarks of denervation-induced muscle atrophy (Fig. 4). Although exact mechanisms remain unknown, it is possible that increased oxidative stress in Tak1-deficient muscle leads to degeneration of NMJs, similar to aging[73]. Indeed, we have previously reported that inactivation of TAK1 induces oxidative stress and mitochondrial dysfunction in skeletal muscle and chronic administration of antioxidants improves muscle mass and contractile function in Tak1[mKO] mice[24].

Our results demonstrate that Smad signaling axis is dramatically dysregulated in skeletal muscle upon inactivation of TAK1 (Fig. 5). There was a remarkable increase in the levels of phosphorylated Smad1/5/8 as well as phosphorylated Smad2 in skeletal muscle upon inactivation of TAK1 in naive conditions. Using co-immunoprecipitation assays, we identified a strong physical interaction between TAK1 and Smad1 proteins in denervated muscle which was less pronounced in the innervated muscle (Fig. 6a, b). The necessity of TAK1 and Smad1 physical interaction was further validated when we observed a more debilitating atrophy upon denervation in Tak1^mKO mice compared to Tak1^fl/fl mice (Fig. 6). Interestingly, we found the levels of phosphorylated Smad1 (p-Smad1) protein are significantly increased in denervated muscle of Tak1^mKO mice compared to Tak1^fl/fl mice. BMP13/BMP14-induced Smad1 activation is known to suppress muscle proteolysis during neurogenic atrophy[1,11,13]. However, our results demonstrate that in the absence of TAK1, a compensatory increase in p-Smad1 levels fails to prevent the activation of components of the ubiquitin-proteasome system. Moreover, TAK1 regulating the spatial distribution of smad proteins could possibly influence the outcome of smad signaling. Although we found a greater nuclear localization of p-Smad1/5/8 in the denervated muscles of Tak1^mKO mice compared to Tak1^fl/fl mice yet it was accompanied with significantly lesser nuclear Smad4 and more nuclear Smad6 levels. In fact, our results suggest that a functional TAK1 facilitates nuclear localization of co-smad, Smad4, and cytosolic retention of inhibitory smad, Smad6, in denervated muscle (Fig. 7). Recent studies have established that Smad2/3 and Smad1/5/8 pathways produce opposing effects on skeletal muscle mass[13,14]. Our experiments found that the phosphorylation of both Smad1/5/8 and Smad2 is enhanced in skeletal muscle during overload-induced hypertrophy and denervation-induced atrophy (Fig. 1a, b, 9g). The reason for Smad2/3 activation during muscle hypertrophy is elusive. Moreover, whether TAK1 can influence the activity and spatial distribution of Smad2/3 are unknown which warrants further investigations.

BMP induced activation of non-canonical Smad signaling through TAK1 has been reported in some experimental models[11,13]. We observed that inactivation of TAK1 in skeletal muscle of mice results in a higher gene expression of *Bmp13* during denervation. Interestingly, our results demonstrate that recombinant BMP7 or BMP13 can trigger phosphorylation of TAK1 along with Smad1/5/8 in cultured myotubes. Moreover, BMP-induced Smad1 activation was greatly hindered when TAK1 was inhibited or knocked down in cultured myotubes (Fig. 8). These results suggest that the exceptional increase in BMP ligands upon denervation may also serve as a mechanism to activate TAK1 which in turn prevents excessive muscle loss by stimulating components of protein translational machinery. Indeed, a role for TAK1 in preventing neurogenic muscle wasting was also evidenced by our experiments demonstrating that the over-expression of TAK1 and TAB1 blunted denervation-induced muscle atrophy (Fig. 9a–f). We also observed that the levels of p-Mnk1 and p-eIF4E, but not p-mTOR, were significantly higher in TAK1/TAB1 overexpressing denervated muscle of mice (Fig. 9g, h) further suggesting that supraphysiological activation of TAK1 inhibits muscle wasting potentially through activation of protein translation machinery.

Cachexia, a devastating multifactorial syndrome observed in advanced stage cancer patients, involves severe muscle wasting[77]. A recent study has demonstrated that NMJ degeneration accompanied with a repression of BMP-Smad1/5/8 signaling precedes the onset of muscle wasting in models of cancer cachexia[78]. Importantly, forced activation of BMP-Smad/1/5/8 pathway using molecular or pharmacological approaches

ameliorates muscle wasting and extends survival in tumor-bearing mice[78]. Our findings revealed a crosstalk between TAK1 and BMP-Smad1/5/8 signaling during neurogenic atrophy which advocates future studies to investigate the involvement of TAK1 in NMJ homeostasis during cancer cachexia and other muscle wasting conditions.

Collectively, our results demonstrate that supraphysiological activation of TAK1 using AAV6-TAK1 and AAV6-TAB1 supports skeletal muscle growth. While more investigations are needed to understand the mechanisms of action of TAK1 in skeletal muscle, our study provides the basis for screening and development of small molecules and peptides that can specifically activate TAK1 to improve skeletal muscle mass and prevent atrophy.

## Methods

**Animals**. C57BL/6 mice were purchased from Jackson Laboratory (Bar Harbor, ME). Skeletal muscle-specific tamoxifen-inducible Tak1-knockout mice (that is, Tak1^mKO) were generated by crossing HSA-MCM mice (The Jackson Laboratory, Tg[ACTA1-cre/Esr1*]2Kesr/J) with floxed Tak1 (that is, Tak1^fl/fl) mice as described[16,23]. All mice were in the C57BL/6 background and their genotype was determined by PCR from tail DNA using AccuStart II PCR genotyping kit (Quantabio, Beverly, MA). The mice were housed in a 12-h light-dark cycle and given water and food ad libitum. Inactivation of TAK1 through Cre-mediated recombination in Tak1^mKO mice was accomplished by injections of tamoxifen (75 mg/kg body weight, i.p.; MilliporeSigma, Burlington, MA, USA) for 4 consecutive days. After a washout period of 4d, the mice were fed a tamoxifen-containing chow (250 mg/kg; Harlan Laboratories, Madison, WI, USA) for the entire duration of the experiment. Littermate Tak1^fl/fl mice (controls) were also treated with tamoxifen and fed tamoxifen-containing chow.

For denervation surgery, mice were anesthetized with isoflurane followed by shaving the right and left hind quarters. Approximately 0.5-cm incision proximal on the lateral side of the right leg was made followed by separating the muscles on the fascia and lifting out the sciatic nerve with a sterile forceps and removing a 2 mm piece of sciatic nerve. Left-side hind limb was subjected to same procedure, except that the sciatic nerve was not transected.

For bilateral SA surgery, the mice were anaesthetized, incision was made proximal to the ankle, and the soleus and ~60% of gastrocnemius were surgically excised from both legs. Finally, the incisions were closed with surgical sutures. A sham surgery was performed in control mice and incisions were closed with surgical sutures. At various time points, hind limb muscles were collected from euthanized mice for biochemical and histology studies.

For local delivery of adeno-associated virus (AAV), mice were anaesthetized with isoflurane and vector genomes ranging from 10^9 to 10^11 (in 30 µl PBS) were injected in the TA muscle of mice. All the animals were handled according to approved institutional animal care and use committee (IACUC) protocols (protocol # PR201900043) of the University of Houston.

**AAV vectors**. AAVs (serotype 6) were custom generated by Vector Biolabs (Malvern, PA, USA). The AAVs expressed either GFP, Tak1 (ref seq #BC006665) or Tab1 (ref seq# BC054369) genes under a ubiquitous CMV promoter.

**Cell culture**. Primary myoblasts were isolated from the hind limb muscles of 8-week old C57BL/6 mice as described previously[79]. The myoblasts were maintained in DMEM/F10 media supplemented with 20% FBS, 1% Pen/Strep and 20 ng/ml basic fibroblast growth factor. The cells were differentiated into myotubes by incubation in DMEM containing 2% horse serum for indicated time periods.

**Generation and use of TAK1 shRNA adenovirus**. Adenoviral vectors expressing either a scrambled shRNA (Ad.Control shRNA) or shRNA targeting mouse Tak1 mRNA (Ad.TAK1 shRNA) was generated following a protocol as described[80]. The target sequences for mouse TAK1 was 5′-GGT GCT GAA CCA TTG CCT TAC-3′. Finally, primary myotubes were transduced with adenoviral vectors at 1:50 multiplicities of infection for 24 h.

**Measurement of protein synthesis**. The rate of protein synthesis in cultured myotubes was measured with a non-isotope-labeled surface sensing of translation (SUnSET) assay as described[80]. In brief, primary myotubes were transduced with AAV6-GFP or a combination of AAV6-TAK1 and AAV6-TAB1 for 24 h, after which cells were washed and incubated in differentiation medium for an additional 24 h. The myotubes were then treated with 1 µM puromycin for exactly 30 min and immediately collected and protein lysates were prepared. Newly synthesized proteins were detected by performing western blot analysis with a primary mouse monoclonal puromycin antibody (clone 12D10, 1:500 dilution; MilliporeSigma).

**Histology and morphometric analysis**. The tibialis anterior (TA) muscle of the mice were isolated, flash frozen in liquid nitrogen, mounted in embedding medium, and sectioned with a microtome cryostat. To assess tissue morphology, 10 μm-thick transverse sections were stained with hematoxylin and eosin (H&E). The sections were examined under an inverted microscope (Eclipse TE 2000-U; Nikon) at room temperature with a Plan × 10/0.25 NA Ph1 DL or a Plan Fluor ELWD × 20/0.45 NA Ph1 DM objective lens, a digital camera (Digital Sight DS-Fi1; Nikon), and NIS Elements BR 3.10 software (Nikon). Images of H&E- or laminin- or dystrophin-stained TA muscle sections were quantified with Nikon NIS Elements BR 3.00 software (Nikon) to measure myofiber cross-sectional area (CSA). For each sample, the frequency distribution of fiber CSA was estimated from about 250 myofibers. Relative amounts of fibrosis in muscle tissues was assayed by using a Masson Trichrome Stain Kit and following a protocol suggested by manufacturer (StatLab, McKinney, TX).

**Immunofluorescence**. For Laminin or dystrophin staining, 10 μm transverse sections of TA muscles were fixed with 4% paraformaldehyde (PFA), blocked with 2% BSA and stained with rabbit anti-Laminin antibody or rabbit anti-dystrophin antibody overnight at 4 °C. Next day, following 3 washes with phosphate buffered saline (PBS), the sections were stained with anti-rabbit Alexa 568 and DAPI.

For cell culture experiments, myotube cultures were fixed with 4% PFA for 15 min at 37 °C, permeabilized with 0.2% Triton-X 100 for 20 min at room temperature and incubated overnight with primary antibodies at 4 °C. Next day, myotubes were incubated with secondary antibodies for 1 h at 37 °C and counterstained with DAPI. The slides were mounted using fluorescence medium (Vector Laboratories) and images were captured using Nikon Eclipse TE 2000-U microscope (Nikon).

For NMJ morphometric analysis, the mice were euthanized and TA muscle isolated was incubated at 4 °C overnight in 30% sucrose solution. The tissues were embedded in OCT and frozen in liquid nitrogen. Next, 30 μm longitudinal sections were cut and fixed with 4% PFA for 10 min. Blocking was done using 2% BSA and mouse on mouse IgG for 1 h. The sections were then incubated with anti-Neurofilament (NF-M) antibody (Clone 2H3, from DSHB) and anti SV2 antibody (from DSHB) for 2 h at room temperature. After washing 3 times with PBS, the sections were incubated with α-Bungarotoxin, Alexa Fluor 488 conjugate, and anti-mouse IgG1-Alexa 568. Nuclei were counterstained with DAPI. The slides were mounted using fluorescence medium (Vector Laboratories) and images were captured using Nikon Eclipse Ti-2e microscope (Nikon) attached with a Prime BSI Express Scientific CMOS (sCMOS) camera.

**Morphometric analysis of NMJs**. For the analysis, NMJ-morph was used which is an ImageJ based standard methodology. The maximum intensity projection images were converted into binary counterparts using ImageJ. Next, a threshold was set using the 'Huang' method. Following the step-by-step user instructions of the NMJ-morph user guide, the morphological variables were analyzed. For example, the basic dimensions such as nerve terminal area and AchR area were measured using standard functions of ImageJ. The terminal nerve endings attach to the nicotinic AchRs located on the surface of the sarcolemma. At this junction, the sarcolemma is folded into depression known as endplate to make space for the nerve endings and the AchRs. The endplate area was estimated by using functions of ImageJ as instructed in the NMJ-morph user guide. Synaptic overlap and synaptic contact area refers to the degree of congruence between the nerve terminals and AchR clusters. The compactness of an NMJ describes the occupied space of the AchR clusters within its endplate. A partially occupied or vacant endplate observed in pathological conditions and is reflected by a decrease in compactness[81]. The AchRs in the endplate remain arranged as pretzel-like fragments. An increase in NMJ fragmentation has been observed in sarcopenia and other disease conditions[82]. NMJ fragmentation was estimated by quantifying the number of discrete AchR clusters. Similarly, pre-synaptic features such as axonal diameter and nerve terminal area was estimated using ImageJ functions. Other pre-synaptic variables such as the average length of branches, number of branch points and terminal branches was derived using NMJ-morph workflow. It uses a formula with a logarithmic derivation, termed complexity to represent the morphological features as a single pre-synaptic index. 20 NMJs from each animal was analyzed for quantification.

**Preparation of cytoplasmic and nuclear extracts**. Briefly, the muscle tissues isolated from mice were immediately frozen in liquid nitrogen and suspended at 1 mg muscle weight per 18 μl of low salt lysis buffer (10 mM HEPES [pH 7.9], 10 mM KCl, 1.5 mM MgCl₂, 0.1 mM EDTA, 0.1 mM EGTA, 1 mM dithiothreitol, 0.5 mM phenylmethylsulfonyl fluoride, and protease inhibitors) followed by mechanical grinding. Cells in the low salt lysis buffer were allowed to swell on ice for ~5 min followed immediately by two cycles of freeze-thaw lysis. The tubes containing the lysed muscle cells were then vortexed vigorously for 10 s, and the lysate was centrifuged for 10 s at 14,000 rpm. The supernatant (cytoplasmic extracts) was removed and saved at −80 °C for further biochemical analysis. The nuclear pellet was resuspended in 4 μl of ice-cold high-salt nuclear extraction buffer (20 mM HEPES [pH 7.9], 420 mM NaCl, 1 mM EDTA, 1 mM EGTA, 25% glycerol, 1 mM dithiothreitol (DTT), 0.5 mM phenylmethylsulfonyl fluoride, and protease

inhibitors) per mg of original muscle weight and was incubated on ice for 30 min with intermittent vortexing. Samples were centrifuged for 5 min at 4 °C, and the supernatant (nuclear extract) was either used immediately or stored at −80 °C.

**Immunoprecipitation and western blotting**. Relative levels of various proteins were determined by performing western blot analysis. Skeletal muscles of mice were washed with sterile PBS and homogenized in lysis buffer consisting of the following: 50 mM Tris-Cl (pH 8.0), 200 mM NaCl, 50 mM NaF, 1 mM DTT, 1 mM sodium orthovanadate, 0.3% IGEPAL, and protease inhibitors. Approximately 100 μg protein was resolved in each lane on 10 to 12% SDS-polyacrylamide gels, electrotransferred onto nitrocellulose membranes, and probed with specific antibodies. The antibodies used are listed in Supplementary Table 1. Bound antibodies were detected by secondary antibodies conjugated to horseradish peroxidase (Cell Signaling Technology). Signal detection was performed by the use of an enhanced chemiluminescence detection reagent (Bio-Rad). Approximate molecular masses were determined by comparison with the migration of prestained protein standards (Bio-Rad). Quantitative estimation of the intensity of each band was performed using ImageJ software (NIH). For loading controls, blots were stripped and reprobed for GAPDH, α-Tubulin or Lamin B1 protein.

For studying protein-protein interaction, immunoprecipitation was performed before immunoblotting. In brief, 700 μg of GA muscle protein extract was incubated overnight at 4 °C with primary antibody against TAK1 or Smad1 in 700 μl of lysis buffer free of DTT. Next day, the antibody–protein complex was collected using Protein A/G Sepharose beads and washed with lysis buffer. The proteins were then run on 10% SDS-polyacrylamide gel electrophoresis gel and immunoblotted with anti-TAK1 or anti-Smad1 antibodies. Uncropped gel images are presented in Supplementary Fig. 9.

**Real-time quantitative PCR (qPCR)**. Total RNA isolated from skeletal muscle tissues of mice or cultured myotubes was subjected to reverse transcription and real-time quantitative PCR (qPCR) analysis, as previously described[22]. The sequence of the primers is described in Supplementary Table 2.

**Statistical analyses and general experimental design**. We calculated sample size using size power analysis methods for a priori determination based on the s.d. and the effect size was previously obtained using the experimental procedures employed in the study. For animal studies, we estimated sample size from the expected number of Tak1^mKO mice and littermate Tak1^fl/fl controls. We calculated the minimal sample size for each group as eight animals. Considering a likely drop-off effect of 10%, we set sample size of each group of six mice. For some experiments, three to four animals were found sufficient to obtain statistical differences. Animals with same sex and same age were employed to minimize physiological variability and to reduce s.d. from mean. The exclusion criteria for animals were established in consultation with IACUC members and experimental outcomes. In case of death, skin injury, sickness or weight loss of >10%, the animal was excluded from analysis. Muscle tissue samples were not used for analysis in cases such as freeze artefacts on histological section or failure in extraction of RNA or protein of suitable quality and quantity. Animals from different breeding cages were included by random allocation to the different experimental groups. Animal experiments were blinded using number codes till the final data analyses were performed. Statistical tests were used as described in the Figure legends. Results are expressed as mean ± SEM. Statistical analyses used two-tailed Student's $t$-test or one-way or two-way ANOVA followed by Tukey's multiple comparison test. A value of $p < 0.05$ was considered statistically significant unless otherwise specified.

**Reporting summary**. Further information on research design is available in the Nature Research Reporting Summary linked to this article.

## Data availability
The authors confirm that the data supporting the findings of this study are available in the supplementary files and source data. All relevant data related to this manuscript are available from the authors upon reasonable request. Source data are provided with this paper.

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

## Acknowledgements

We thank S. Akira (Osaka University, Osaka, Japan) for providing floxed TAK1 mice. This work was supported by funding from NIH grants AR059810 and AR068313 (to A.K.).

## Author contributions

A.R. and A.K. designed the work. A.R. performed all the experiments and analyzed results. A.R. wrote the first draft of the manuscript. A.K. edited and finalized the manuscript.

## Competing interests

The authors declare no competing interests.
