## [Peer Review File · Nature Communications]

Reviewers' Comments:

Reviewer #1:

Remarks to the Author:

The present work aims to dissect the insights that link TAK1 to muscle mass regulation. Starting from the previously published findings that TAK1 ablation induces muscle loss, the authors overexpressed TAK1 and showed that muscle growth was induced. This hypertrophy was associated to an increased phosphorylation of Smad1/5/8, SMAD2, p38MAPK, ERK1/2, NFkB, rpS6, p90S6K and several initiation factors of translation but not of AKT, mTOR and p70S6K. Interestingly, they also found that inhibition of TAK1 causes dismantle of neuromuscular junction (NMJ) and denervation features that can explain the exacerbated muscle loss and the weakness of the knockout mice. Absence of TAK1 caused an exacerbated muscle loss after denervation while TAK1 overexpression partially blunted denervation-induced muscle loss. Mechanistically, the authors found that TAK1 interacts with Smad1 and control the nuclear localization of Smad4. The connection between TAK1 with Smad1 and protein synthesis is novel and therefore, of great interest for a broad audience. However, some of the mechanistic insights should be better characterized. The authors should consider the following points.

Point1. Authors should better dissect how TAK1 promotes protein synthesis and hypertrophy. They should use at least in myotubes whether rapamycin treatment or knockdown of Smad1-5-9 blunt puromycin incorporation and myotube growth. This is important to exclude mTOR and support a role of Smad1/5/8 on protein synthesis. Also, similar to the in vivo experiment it will important to monitor the phosphorylation status of p-Smad1/5/9, p-mTOR, p-ERK, p-Mnk1, p-P90RSK1

Point2. The author found that TAK1 interacts with Smad1 but only Smad4 nuclear localization is affected by TAK1 inhibition. Does TAK1 also bind to Smad4? Is it an assembly site for Smad1-Smad4 complex formation?

Point3. Figure 5E. To be consistent with Fig1 and Fig.2, the quantification of the western blots should be shown as single p-Smad1 and not as ratio of p-Smad1/Smad1. Indeed, it is important to underline that the best way to express values, is to normalize the phospho-Smad1 and phospho-Smad3 protein levels relative to a stable housekeeping protein. This is essential when the phosphorylation activates the protein. In fact, if the total protein as well as its phosphorylation status increase, targets of the pathway would be strongly activated despite the ratio of phosphor-Smad1 vs. total Smad1 is not changing. The opposite is true in the case of a downregulation of the total protein and of its phosphorylation. Therefore, the right approach is to normalize the phospho-protein for a reliable housekeeping protein. Conversely, when the phosphorylation inhibits the protein, then the ratio of the phosphor on the total is a must. In fact, an increase of total protein associated to a maintenance of the phosphorylation result in a strong inhibitory action (e.g. eIF2a)

Point4. Figure6A: please show GAPDH to confirm that the nuclear fraction is pure. Figure 6B: please show the LaminB to confirm that the cytosolic fraction is pure.

Point5. Figure7. Please show puromycin incorporation after BMP7/13 treatment in presence or absence of 5-Z7O or, alternatively, after shRNA treatment. This experiment will confirm whether BMP-TAK1 pathway is involved in the increase of protein synthesis. The text of page14 line 414 "suggest that TAK1 mediates" should be modified in "suggest that TAK1 contributes" because the phosphorylation of Smads is not completely blunted.

Point6. Figure 8. The authors showed the effect of TAK1 overexpression 7 days after denervation when atrophy is mild. What about at 14 days after denervation when 40-50 % of muscle mass is lost? Also, it would be an interesting addition to show the effect on other muscles such as soleus (rich in type 1 fibers) and gastro

Point7. Figure 8G. Quantification of western blots should show the single phospho- or tota- protein (see point3) for the activating phosphorylation sites.

Point8. Figure8. The graph of CSA distribution should include the overlap between control and

denervated in flox and KO mice

Minor Point: Page 9 line 274. Citation of the sentence "FoxOs controlling autophagy and ubiquitin proteasome" is missing. Please include Milan et al 2015 Nat Comm. Where ChIP and loss of function experiments were performed.

Reviewer #2:

Remarks to the Author:

Although the importance of TAK-1 in muscle growth is already reported in conditional ablation studies, the authors used a different set of studies using conditional TAK1 ablation. The current study also used TAK1 forced activation and overexpression experiments. Although the TAK1 activation is supraphysiological, the study will be of some help for therapeutic strategy for neurogenic atrophy. However, introduction of two molecules (TAK1 and TAB1) would not be a good therapeutic strategy; therefore, it is advisable to state the potential therapy deduced from the current study.

As the authors mentioned in the discussion, there are several literatures showing that the TAK-1 inhibition, but not activation, prevents muscle wasting (ref 63-65). The authors did discuss the difference; however, it is not convincing enough for the readers. More rational reason and scientific support for TAK1 to mitigate atrophy are necessary.

Comments are as follows.

The reviewer understands that TAB1 is needed for TAK1, and the authors used cotransfection. As a control, it is advisable to add experiments using AAV6-TAK1 only and AAV6-TAB1 only in at least some key experiments.

In Fig.5, the reviewer understands the interaction of TAK1 and Smad1, and changes of E3 ligases in Tak1(mKO). However, the relationship of Smad1 and E3 ligases are not studied. This weakens the importance of the study.

Figure 6 data is descriptive, but not mechanistic. In Tak1(mKO), p-Smad1/5/9 is high in the nucleus, whereas the amount of their partner co-Smad, Smad4, is reduced in denervation. Does the BMP/Smad1/5/9 activity actually activated? The reduction of the existence of Smad4 in the nucleus in Tak1(mKO) does not directly show that TAK1 supports the nuclear translocation of Smad4 in the nucleus, since it is the inhibition experiment.

In Fig.2C, what are the two bands seen in eIF4H lane?

What is the evidence of heterotrimerization of Smad1/5/9 with Smad4? Does the heterotrimerization mean Smad1/5/4, Smad1/8/4 or Smad5/8/4?

In Fig. 4A-B, given that both p-Smad2 and Smad2 are elevated in Tak1(mKO). The ratio of pSmad2/Smad2 should be shown.

The authors measured mRNAs for BMP and TGFbeta ligands. TGF-beta family ligands form homo- and heterodimers and undergo processing for activation as peptides. The level of mRNA does not always reflect mature protein levels and their activities.

Other comments

Please correct and check following errors and typos.

Unify either Smad1/5/8 or Smad1/5/9. Smad1/5/8 and Smad1/5/9 appeared rather randomly in the text. In Figures, the authors used Smad1/5/9.

Line 120, delete extra period before ref 17, 19.

Line 177, it is better to add AAV6 before TAK1 and TAB1.

Line 315, Activin ligands are vague. Is it inhibin betaA coding for activin A ligand?

Lines 337-339 in page 11, cite Fig. 5C.

Line 413, Fig.7G should be Fig.7F.

Line 684, DTT is already appeared before. Abbreviate it when it first appeared.

Fig.1 legend.

m-TOR would be mTOR. (H) should read (F).

In Figure S1 and S2 legend, unify μm or μm .

In Figure 3C legend, line 973: add period before (C).

Reviewer #3:

Remarks to the Author:

Roy and Kumar present an interesting manuscript on the role of TAK1 in promoting skeletal muscle mass. Through extensive *in vitro* and *in vivo* experiments, they rather convincingly identify an important role for TAK1 in supporting muscle growth and restricting atrophy in response to nerve transection. TAK1 and associated pathways are shown to be activated in response to muscle overloading, while AAV-mediated TAK1 overexpression results in myofiber hypertrophy, which is associated with activation of the protein synthesis machinery. Muscle-specific reduction in TAK1 is reported to impair integrity of the neuromuscular junction and disrupt Smad signalling, while TAK1-mediated Smad signalling appears to be important for alleviating the severity of muscle atrophy in response to denervation. Finally, TAK1 overexpression was shown to limit atrophy caused by experimental denervation.

Overall, the manuscript is well written, logical and clearly signposted, which is particularly important given the density and complexity of the experimental findings. Furthermore, the data appear robust and coherent across experimental platforms. Moreover, most of the experiments incorporate data from large sample sizes, the figures are clearly presented with helpful colour-schemes that run throughout the manuscript, and the results are nicely discussed and placed within the field.

Nonetheless, I have several major comments and minor suggestions that the authors should address before the work is considered for publication.

Major

1. The weakest part of the manuscript is the NMJ analysis. Assessment of the neuronal component of an NMJ classically relies upon an antibody combination of 2H3-neurofilament and a pre-synaptic marker (e.g., SV2 or synaptophysin). By not using a pre-synaptic marker, the authors are likely to be missing a major anatomical part of the NMJ, negatively impacting the pre-synaptic morphological analysis. This is confirmed by the small degree of synaptic overlap observed in control mice (Figure 3C) – I would expect to see more like 65-85%.
2. Perhaps related to the above point, how do the authors explain that there is no difference in nerve terminal area between control and Tak1 mKO, but there is a reduction in the percentage of synaptic overlap? Larger endplates could explain this, but they are in fact smaller in the Tak1 mutants. Similarly, can the authors please speculate how Tak1 knockout in muscle would cause the pre-synaptic phenotypes presented in Figure 3C.
3. The rationale behind co-administering AAV6-TAK1 with AAV6-TAB1 is explained based on the literature (Refs 18 and 26). However, I feel it would be particularly informative to present preliminary representative results of experiments performed in mice receiving single AAVs (i.e., AAV6-TAK1 or AAV6-TAB1). This would confirm the assertion that, "TAK1 has no kinase activity when ectopically expressed alone but is activated when co-expressed with TAB1"; this will validate the chosen dual-AAV approach, and confirm that the assertion holds in this experimental setting.
4. Relating to the previous point, TAB1 overexpression upon AAV6-TAB1 treatment is not confirmed at any point and should be.
5. AAV-mediated expression of GFP and TAB1 are also not confirmed in the cultured myotubes and should be.
6. Puromycin levels should really be quantified to confirm the increased translation in the TAK1/TAB1 overexpressing cultures. The ubiquitination experiments in Figure 5 should also be quantified and repeated an appropriate number of times,
7. Were the immunoprecipitation experiments repeated and/or quantified? Doing so will perhaps allow the authors to see whether the interaction between TAK1 and Smad1 is indeed greater upon

denervation, as is hinted at in the text.

8. In earlier figures (1-4), phosphorylated forms of proteins are quantified relative to the loading control GAPDH; however, in Figures 5 and 8, phosphorylated protein levels are calculated relative to total forms of respective proteins. To aid interpretation, both methods should be applied to all relevant data, or perhaps one method used throughout.

Minor

1. In the abstract, it is not clear what is meant by "proximal signalling" – please clarify.
2. The purpose (e.g., skeletal muscle growth) and experimental detail (e.g., muscles removed) of the bilateral synergistic ablation surgery should be better explained on page 5 when first introduced.
3. Line 180: TAB1 should also be included in this final sentence of the section.
4. The extent of Tak1 knockdown upon tamoxifen treatment should really have been shown in both the tibialis anterior (TA) and gastrocnemius (GA) muscles analysed, as opposed to just the GA.
5. To describe the NMJ as having three components, is correct, but only two of the three mentioned are classically described in the tripartite NMJ. The authors should probably mention that terminal Schwann cells are also a critical player at the neuromuscular synapse if they are going to introduce the NMJ in this way.
6. Features assessed using NMJ morph need to be better introduced for those not familiar with NMJs and the software. For instance, how do AChR and endplate areas differ? What are compactness and fragmentation?
7. Please include details of how many NMJs per animal were assessed for the morphological analyses – I cannot seem to find the detail.
8. There appears to be an issue with the NMJ image presented for the control mice (top row, Figure 3A) – the red and blue channels appear shifted to the left and out of sync with the green channel. Also, red/green immunofluorescent colour-combinations should be avoided due to issues for certain forms of colour-blindness – please pseudocolour.
9. Reference to Fig. 5C is missing from the text (lines 337-339).
10. The title of Figure S5 contradicts the findings presented in Figure 1E-G.

RESPONSE TO REVIEWERS' COMMENTS

We sincerely thank all the three reviewers for their time and efforts for reviewing our manuscript. We are also grateful to the reviewers for finding our manuscript interesting and providing constructive suggestions. Major changes made in the manuscript are highlighted by yellow background. Our pointwise response to reviewers' comments is as follows:

REVIEWER #1

The present work aims to dissect the insights that link TAK1 to muscle mass regulation. Starting from the previously published findings that TAK1 ablation induces muscle loss, the authors overexpressed TAK1 and showed that muscle growth was induced. This hypertrophy was associated to an increased phosphorylation of Smad1/5/8, SMAD2, p38MAPK, ERK1/2, NF- κ B, rpS6, p90S6K and several initiation factors of translation but not of AKT, mTOR and p70S6K. Interestingly, they also found that inhibition of TAK1 causes dismantle of neuromuscular junction (NMJ) and denervation features that can explain the exacerbated muscle loss and the weakness of the knockout mice. Absence of TAK1 caused an exacerbated muscle loss after denervation while TAK1 overexpression partially blunted denervation-induced muscle loss. Mechanistically, the authors found that TAK1 interacts with Smad1 and control the nuclear localization of Smad4. The connection between TAK1 with Smad1 and protein synthesis is novel and therefore, of great interest for a broad audience. However, some of the mechanistic insights should be better characterized. The authors should consider the following points.

OUR RESPONSE: We thank the reviewer for finding our work novel and interesting. Our response to reviewer's comments is as follows:

Point1. Authors should better dissect how TAK1 promotes protein synthesis and hypertrophy. They should use at least in myotubes whether rapamycin treatment or knockdown of Smad1-5-9 blunt puromycin incorporation and myotube growth. This is important to exclude mTOR and support a role of Smad1/5/8 on protein synthesis. Also, similar to the in vivo experiment it will important to monitor the phosphorylation status of p-Smad1/5/9, p-mTOR, p-ERK, p-Mnk1, p-P90RSK1.

OUR RESPONSE: We thank the reviewer for this valuable suggestion. We have now performed additional experiments. Our new results demonstrate that similar to our findings in skeletal muscle of mice, the levels of p-ERK1/2, p-Mnk1, p-rpRS6, and p-eIF4E were significantly increased in cultured mouse primary myotubes transduced with AAV6-TAK1 and AAV6-TAB1. Our results also showed that there was no significant difference in the levels of p-mTOR and p-Smad1/5/8 between control and TAK1 and TAB1 overexpressing cultures (please refer to new Figure 3b, c, description on Page # #8-9, highlighted text). We also performed another experiment to determine whether forced activation of TAK1 using AAV6-TAK1 and AAV6-TAB1 can rescue protein synthesis in the presence of rapamycin, a well-known inhibitor of mTOR. Our results showed that overexpression of TAK1 and TAB1 significantly improved the rate of protein synthesis in rapamycin-treated cultured myotubes (new Figure 3f, S3g and description of page # 9, highlighted text). These results suggest that the activation of TAK1 stimulates translational

machinery and protein synthesis independent of activation of mTOR. We also studied the effect

of knockdown of Smad1 on protein synthesis in myotubes.

Surprisingly, we found that instead of inhibition, siRNA-mediated knockdown of Smad1, improves protein synthesis in cultured myotubes in naïve conditions. A representative gel from two independent experiments is presented here. We did not find any changes in phosphorylation of mTOR upon knockdown of Smad1. The activation of protein synthesis upon knockdown of Smad1 may be a compensatory mechanism to prevent atrophy in cultured myotubes. We have also mentioned in the following sections and emphasize here too that our experiments suggest that TAK1

stimulates translational machinery through stimulating P90RSK--Mnk1/2-eIF4E axis. However, TAK1 appears to be inhibiting neurogenic muscle atrophy through the activation of Smad1/5/8 pathway.

Point2. The author found that TAK1 interacts with Smad1 but only Smad4 nuclear localization is affected by TAK1 inhibition. Does TAK1 also bind to Smad4? Is it an assembly site for Smad1-Smad4 complex formation?

OUR RESPONSE: We performed multiple co-immunoprecipitation experiments to investigate whether TAK1 directly interacts with Smad4 in denervated skeletal muscle of mice. For this experiment, we immunoprecipitated innervated and denervated muscle extracts using anti-Smad4 antibody followed by immunoblotting with anti-TAK1 antibody. However, we did not find any enrichment of TAK1. We have mentioned this in the manuscript and included results as supplementary figure in our revised manuscript (Supplemental Figure S7a). In future studies, we will perform additional experiments to understand how TAK1 affects the nuclear localization of Smad4 in denervated skeletal muscle.

Point3. Figure 5E. To be consistent with Fig1 and Fig.2, the quantification of the western blots should be shown as single p-Smad1 and not as ratio of p-Smad1/Smad1. Indeed, it is important to underline that the best way to express values, is to normalize the phospho-Smad1 and phospho-Smad3 protein levels relative to a stable housekeeping protein. This is essential when the phosphorylation activates the protein. In fact, if the total protein as well as its phosphorylation status increase, targets of the pathway would be strongly activated despite the ratio of phosphor-Smad1 vs. total Smad1 is not changing. The opposite is true in the case of a downregulation of the total protein and of its phosphorylation. Therefore, the right approach is to normalize the phospho-protein for a reliable housekeeping protein. Conversely, when the phosphorylation inhibits the protein, then the ratio of the phosphor on the total is a must. In fact, an increase of total protein associated to a maintenance of the phosphorylation result in a strong inhibitory action (e.g. eIF2a)

OUR RESPONSE: We completely agree with the reviewer. We have now provided quantification where the levels of phosphorylated protein is normalized with a housekeeping protein. We have removed the ratio of phosphorylated vs total protein throughout the manuscript.

Point4. Figure 6A: please show GAPDH to confirm that the nuclear fraction is pure. Figure 6B: please show the LaminB to confirm that the cytosolic fraction is pure.

OUR RESPONSE: We have now performed western blots for GAPDH and Lamin B1 to show purity of the nuclear and cytoplasmic fractions. Our experiments show that Lamin B1 is predominately present in nuclear extracts only. Since GAPDH can be present in both cytoplasm and nucleus, we found some but reduced levels of GAPDH in nuclear extracts compared with cytoplasmic extracts. Please refer to Figure 6a and 6b for these results and text on page #11.

Point5. Figure7. Please show puromycin incorporation after BMP7/13 treatment in presence or absence of 5-Z70 or, alternatively, after shRNA treatment. This experiment will confirm whether BMP-TAK1 pathway is involved in the increase of protein synthesis. The text of page14 lane 414 “suggest that TAK1 mediates” should be modified in “suggest that TAK1 contributes” because the phosphorylation of Smads is not completely blunted.

OUR RESPONSE: We agree with the reviewer and therefore have now mentioned that TAK1 contributes to the BMP-induced Smad1 signaling in skeletal muscle.

We performed additional experiments to study puromycin incorporation after BMP7 or BMP13 treatment. However, we could not find any up-regulation of protein synthesis in cultured myotubes after treatment with either BMP7 or BMP13. Specifically, we transduced cultured myotubes with Ad.Control or Ad.TAK1 shRNA. After 48h, the myotubes were treated with PBS, BMP7, or BMP13 for additional 4h. During the last 30 min of incubation, puromycin (1 μ M) was added to the myotubes. Western blot analysis showed a significant decrease in protein

synthesis in cultured myotubes transduced with Ad.TAK1 shRNA compared to control myotubes transduced with Ad.Control shRNA. However, we did not find any additional increase in protein synthesis with 4h treatment with BMP7 or BM13 compared with PBS treatment in Ad.Control shRNA transduced myotubes (please see a representative immunoblot here). We believe that BMP-mediated activation of TAK1 and Smad1, as well as their interaction are involved in regulating protein ubiquitination and degradation during denervation. This is

also evident from our previous observation that inducible inactivation of TAK1 leads to increased expression of MuRF1, MAFbx and MUSA1 with consequently more protein ubiquitination (Hindi et al, JCI Insight. 2018 Feb 8;3(3):e98441). Furthermore, in our present study, we found that the levels of MAFbx, MuRF1, and MUSA1 as well as polyubiquitination was significantly increased in denervated muscle of TAK1^{mkO} mice compared to Tak1^{fl/fl} mice (Fig. 6d-g). Our results using cultured myotubes demonstrate that knockdown or pharmacological inhibition of TAK1 attenuates the BMP-induced phosphorylation of Smad1/5/8 which further suggests a role for TAK1 in supporting the activity of Smad1 (Fig. 8).

Point6. Figure 8. The authors showed the effect of TAK1 overexpression 7 days after denervation when atrophy is mild. What about at 14 days after denervation when 40-50 % of

muscle mass is lost? Also, it would be an interesting addition to show the effect on other muscles such as soleus (rich in type 1 fibers) and gastro

OUR RESPONSE: We thank the reviewer for this important suggestion. We have now performed an additional experiment to study the effect of overexpression of TAK1 and TAB1 on muscle atrophy at day 14 after denervation. Our new results demonstrate that overexpression of TAK1 and TAB1 significantly reduced the denervation-induced loss of myofiber CSA in TA muscle of mice after 14 days of denervation (Supplemental Fig. S8). Description on Page # 16, highlighted text. We have also performed an additional experiment to investigate whether TAK1 can improve myofiber growth in gastrocnemius muscle. Consistent with our results with TA muscle, we found that the forced activation of TAK1 through overexpression of TAK1 and TAB1 causes myofiber hypertrophy in GA muscle of wild-type mice (Supplemental Fig. S2).

Point7. Figure 8G. Quantification of western blots should show the single phospho- or total-protein (see point3) for the activating phosphorylation sites.

OUR RESPONSE: In the revised manuscript, we now show levels of phosphorylated and total protein separately for all the experiments in the manuscript.

Point8. Figure8. The graph of CSA distribution should include the overlap between control and denervated in flox and KO mice.

OUR RESPONSE: We have now provided overlapping relative frequency distribution plots of control and denervated muscle in $Tak1^{fl/fl}$ and $Tak1^{mKO}$ mice (Revised Figs 8e and 8f).

Minor Point: Page 9 line 274. Citation of the sentence “FoxOs controlling autophagy and ubiquitin proteasome” is missing. Please include Milan et al 2015 Nat Comm. Where CHIP and loss of function experiments were performed.

OUR RESPONSE: This has been added as Reference # 44. Thank you!

REVIEWER #2

Although the importance of TAK-1 in muscle growth is already reported in conditional ablation studies, the authors used a different set of studies using conditional TAK1 ablation. The current study also used TAK1 forced activation and overexpression experiments. Although the TAK1 activation is supraphysiological, the study will be of some help for therapeutic strategy for neurogenic atrophy. However, introduction of two molecules (TAK1 and TAB1) would not be a good therapeutic strategy; therefore, it is advisable to state the potential therapy deduced from the current study.

OUR RESPONSE: Our present study was designed to obtain initial evidence on how activation of TAK1 using molecular approaches affects skeletal muscle mass. Our experimentation clearly suggests that forced activation of TAK1 in myofibers induces muscle growth. We have used

combination of TAK1 and TAB1 which is an established method to achieve supraphysiological activation of TAK1 in mammalian cells. We agree that the actual therapy may be quite different which may involve use of small molecules and/or expression of other regulators/peptides that can specifically activate TAK1 in skeletal muscle. This has been mentioned in the last paragraph of the “Discussion” section of the manuscript (page # 21, highlighted text).

As the authors mentioned in the discussion, there are several literatures showing that the TAK-1 inhibition, but not activation, prevents muscle wasting (ref 63-65). The authors did discuss the difference; however, it is not convincing enough for the readers. More rational reason and scientific support for TAK1 to mitigate atrophy are necessary.

OUR RESPONSE: There are studies (ref 63-65) which show that the inhibition of TAK1 prevents muscle wasting. However, those studies tested the effect of TAK1 inhibition in chronic disease states where TAK1 may have been inhibited not only myofibers but also in other cell types, including inflammatory immune cells that are present in muscle microenvironment. We have used a targeted approach in which TAK1 is specifically activated in skeletal muscle of wild-type mice. The role of TAK1 could be quite distinct in different cell types and disease conditions. The results of our present study demonstrate that forced activation of TAK1 causes myofiber hypertrophy in naïve condition. Moreover, overexpression of TAK1 and TAB1 prevents denervation-induced myofiber atrophy in mice. We have now performed some additional experiments to further support the role of TAK1 in mitigating muscle atrophy. Our results indicate that the activation of TAK1 observed in overload induced muscle hypertrophy and also in denervation induced atrophy has a growth promoting role. We also improved the “Discussion” section to include reviewer’s point (Page # 18, last paragraph highlighted text).

Comments are as follows.

The reviewer understands that TAB1 is needed for TAK1, and the authors used cotransfection. As a control, it is advisable to add experiments using AAV6-TAK1 only and AAV6-TAB1 only in at least some key experiments.

OUR RESPONSE: We thank the reviewer for this suggestion. We have now performed additional experiments in which we also employed AAV6-TAK1 or AAV6-TAB1 alone. As our data in Figure 1 shows that co-injection of AAV-TAK1 and AAV-TAB1, but not AAV-TAK1 or AAV6-TAB1 alone induces myofiber hypertrophy in adult mice. More importantly, our new results clearly demonstrate that the levels of phosphorylated TAK1 are increased only when AAV6-TAK1 was injected along with AAV-TAB1 (Fig. 1g). Furthermore, our new results suggest that co-expression of TAK1 and TAB1 is essential for stimulating translational machinery and protein synthesis in skeletal muscle of mice (Supplemental Fig. S3). Collectively, these experiments confirm that co-expression of TAK1 and TAB1 is essential for the activation of TAK1 and downstream signaling in skeletal muscle.

In Fig.5, the reviewer understands the interaction of TAK1 and Smad1, and changes of E3 ligases in Tak1(mKO). However, the relationship of Smad1 and E3 ligases are not studied. This weakens

the importance of the study.

OUR RESPONSE: In the present study, we demonstrate a novel interaction between TAK1 and Smad1 in the denervated skeletal muscle. The role of BMP-Smad1/5/8 pathway in the regulation of muscle-specific E3 ubiquitin ligases has been extensively investigated in the published reports (e.g. Winbanks et al, J Cell Biol. 2013 Oct 28; 203(2): 345–357; Sartori et al. Nat Genet. 2013 Nov;45(11):1309-18; Winbanks et al. Sci Transl Med. 2016 Jul 20;8(348):348ra98.). We had cited those published studies in our manuscript (ref # 10-15).

Figure 6 data is descriptive, but not mechanistic. In Tak1(mKO), p-Smad1/5/9 is high in the nucleus, show that TAK1 supports the nuclear translocation of Smad4 in the nucleus, since it is the inhibition experiment.

OUR RESPONSE: In the context of denervation-induced muscle atrophy, phosphorylated Smad1 is known to form complex with Smad4 and translocate to nucleus to limit expression of atrogenes (Sartori et al. Nat Genet. 2013 Nov;45(11):1309-18). We found an increase in p-Smad1/5/8 in nuclear fraction in denervated muscle which is consistent with previously published reports and the notion that activation of BMP-Smad1/5/8 pathway is a mechanism to prevent excessive muscle loss during denervation. We found that the levels of p-Smad1/5/8 was higher in both cytoplasmic and nuclear fractions of denervated muscle of Tak1^{mKO} mice compared to denervated muscle of Tak1^{fl/fl} mice. Our experiments also show that inactivation of TAK1 leads to a significant reduction in nuclear Smad4 levels in denervated skeletal muscle of Tak1^{mKO} compared to corresponding denervated muscle of Tak1^{fl/fl} mice. In addition, we observed a significant increase in the levels of nuclear Smad6 (an inhibitor of Smad1/5/8) in denervated muscle of Tak1^{mKO} mice. The results suggest that TAK1 is involved in the spatial regulation of Smad proteins in denervated muscle. We agree that further investigations are needed to understand how TAK1 supports the nuclear translocation of Smad4 and cytoplasmic retention of Smad6 in denervated muscle. Therefore, we have changed the sentence in our result section as follows: “Our results suggest that TAK1 regulates the spatial distribution of Smad1, Smad4 and Smad6 in denervated muscle. However, further investigations are needed to understand how TAK1 supports the nuclear translocation of Smad4 and cytosolic retention of Smad6 protein in denervated muscle.” (Page # 14, highlighted text). Additionally, we have changed the Result subheading as “TAK1 regulates subcellular distribution of Smad proteins in denervated skeletal muscle.”

In Fig.2C, what are the two bands seen in eIF4H lane?

OUR RESPONSE: eIF4H is an elongation initiation factor essential for functioning of the eIF4A helicase in the eIF4F complex. As a result of alternate splicing, it is expressed as two protein isoforms of 25 and 27 kDa. The anti- eIF4H antibody from Cell Signaling Technology (Catalog #2444) detects both bands.

What is the evidence of heterotrimerization of Smad1/5/9 with Smad4? Does the heterotrimerization mean Smad1/5/4, Smad1/8/4 or Smad5/8/4?

OUR RESPONSE. We have replaced the word heterotrimerization with multiprotein complex in our revised manuscript. The Smad proteins are structurally similar proteins that mediate signal transduction for the receptors of the TGF- β superfamily. There are three distinct sub-types of Smad proteins, the R-smads (receptor regulated smads), Co-Smads (common partner Smads) and I-smads (inhibitory smads). R-Smads include Smad2 and Smad3 from the TGF- β /Activin/Nodal branch and Smad1, Smad5 and Smad8 (which is also known as Smad9) from the BMP/GDF branch of TGF- β signaling. Depending upon the ligands, the R-smads (Smad2 and Smad3 or Smad1 and Smad5 and Smad 8) are activated. Once activated the R-smads typically bind to common mediator Smad4 and form oligomers. It is thought that the Smad oligomer is typically made up of two R-smads and a co-smad (Smad2-Smad2-Smad4 complex or Smad2-Smad3-Smad4 complex) which then translocate to the nucleus. The inhibitory smads (Smad6 and Smad7) compete with Smad4 to bind to the R-Smad dimer to form a trimer and block their ability to transcribe genes.

In Fig. 4A-B, given that both p-Smad2 and Smad2 are elevated in Tak1(mKO). The ratio of pSmad2/Smad2 should be shown.

OUR RESPONSE: As suggested by the Reviewer # 1, it is most appropriate to show fold change in phosphorylated and total protein separately after normalizing with a house keeping protein. Therefore, we have now presented quantification of all the phosphorylated and total protein levels separately.

The authors measured mRNAs for BMP and TGFbeta ligands. TGF-beta family ligands form homo- and heterodimers and undergo processing for activation as peptides. The level of mRNA does not always reflect mature protein levels and their activities.

OUR RESPONSE: The pattern of upregulated BMPs and GDFs upon denervation is well-known and reported in published studies. We do understand that mRNA levels do not necessarily reflect their protein expression levels. However, here in our report we studied the differential expression of genes in TAK1^{fl/fl} and TAK1^{mKO} mice and matched them to the expression patterns observed in denervated and innervated muscle. Our results showing an increase in mRNA levels of BMPs and GDFs, along with changes in phospho-Smad levels and up-regulation of Foxo proteins and HDAC4-myogenin axis in TAK1^{mKO} mice compared with TAK1^{f/f} mice indicate that muscle atrophy due to Tak1 inactivation mimicked neurogenic atrophy phenotype. This interested us to study the NMJ morphology and NMJ gene expression which showed a similar pattern as observed in skeletal muscle in response to denervation.

Other comments:

Please correct and check following errors and typos.

Unify either Smad1/5/8 or Smad1/5/9. Smad1/5/8 and Smad1/5/9 appeared rather randomly in the text. In Figures, the authors used Smad1/5/9.

OUR RESPONSE: Because all published studies used Smad1/5/8 for denervated muscle, we have decided to use Smad1/5/8 instead of Smad1/5/9 in our manuscript. We have made

necessary changes throughout the manuscript.

Line 120, delete extra period before ref 17, 19.

Done

Line 177, it is better to add AAV6 before TAK1 and TAB1.

Done

Line 315, Activin ligands are vague. Is it inhibin betaA coding for activin A ligand?

OUR RESPONSE: Yes. It was Inhibin beta A. we have made this correction.

Lines 337-339 in page 11, cite Fig. 5C.

OUR RESPONSE: This figure has been moved to Supplemental Data file. It has been referred.

Line 413, Fig.7G should be Fig.7F.

OUR RESPONSE: This has been fixed.

Line 684, DTT is already appeared before. Abbreviate it when it first appeared.

OUR RESPONSE: This has been done.

Fig.1 legend.

m-TOR would be mTOR. (H) should read (F).

OUR RESPONSE: This has been corrected.

In Figure S1 and S2 legend, unify um or μm .

OUR RESPONSE: We have now used μm .

In Figure 3C legend, line 973: add period before (C).

OUR RESPONSE: Done

REVIEWER #3

Roy and Kumar present an interesting manuscript on the role of TAK1 in promoting skeletal muscle mass. Through extensive in vitro and in vivo experiments, they rather convincingly identify an important role for TAK1 in supporting muscle growth and restricting atrophy in response to nerve transection. TAK1 and associated pathways are shown to be activated in response to muscle overloading, while AAV-mediated TAK1 overexpression results in myofiber hypertrophy, which is associated with activation of the protein synthesis machinery. Muscle-specific reduction in TAK1 is reported to impair integrity of the neuromuscular junction and disrupt Smad signalling, while TAK1-mediated Smad signalling appears to be important for alleviating the severity of muscle atrophy in response to denervation. Finally, TAK1 overexpression was shown to limit atrophy caused by experimental denervation.

Overall, the manuscript is well written, logical and clearly signposted, which is particularly important given the density and complexity of the experimental findings. Furthermore, the data appear robust and coherent across experimental platforms. Moreover, most of the experiments incorporate data from large sample sizes, the figures are clearly presented with helpful colour-schemes that run throughout the manuscript, and the results are nicely discussed and placed within the field.

Nonetheless, I have several major comments and minor suggestions that the authors should address before the work is considered for publication.

OUR RESPONSE: We thank the reviewer for finding our manuscript interesting. We have now addressed and incorporated all the suggestions by the reviewer.

Major

1. The weakest part of the manuscript is the NMJ analysis. Assessment of the neuronal component of an NMJ classically relies upon an antibody combination of 2H3-neurofilament and a pre-synaptic marker (e.g., SV2 or synaptophysin). By not using a pre-synaptic marker, the authors are likely to be missing a major anatomical part of the NMJ, negatively impacting the pre-synaptic morphological analysis. This is confirmed by the small degree of synaptic overlap observed in control mice (Figure 3C) – I would expect to see more like 65-85%.

OUR RESPONSE: We sincerely thank the reviewer for this important suggestion. We have now repeated the immunofluorescence staining of NMJs from TA muscle of $Tak1^{fl/fl}$ and $Tak1^{mKO}$ mice with combination of 2H3 and SV2 (please see now Figure 4a). We found a significant decrease in AChR area, end plate area, percentage of synaptic overlap, synaptic contact area, and percent compactness in $TAK1^{mKO}$ mice compared with $TAK1^{fl/fl}$ mice (now Figure 4a and 4b). From our new analysis, we found the mean percent synaptic overlap in $Tak1^{fl/fl}$ mice was 73.18 ± 2.35 whereas in $TAK1^{mKO}$ mice it was 62.77 ± 2.44 ($n=3$ in each group).

2. Perhaps related to the above point, how do the authors explain that there is no difference in nerve terminal area between control and $Tak1^{mKO}$, but there is a reduction in the percentage of synaptic overlap? Larger endplates could explain this, but they are in fact smaller in the $Tak1$ mutants. Similarly, can the authors please speculate how $Tak1$ knockout in muscle would cause the pre-synaptic phenotypes presented in Figure 3C.

OUR RESPONSE: From our analysis, we found that there were severe defects in the NMJs of $Tak1^{mKO}$ mice compared with $Tak1^{fl/fl}$ mice. Some of the endplates in the knockout mice were oddly smaller whereas some larger but with extensive fragmentation (Supplemental Fig. S4c and S4d). We have now included the representative images in our revised manuscript (Fig. S4d). Moreover, we observed some decrease, but not statistically significant ($P=0.1$) in nerve terminal area in the NMJs of $Tak1^{mKO}$ mice compared with $Tak1^{fl/fl}$ mice. The $Tak1^{mKO}$ mice undergoes dramatic muscle wasting and the gene expression patterns in muscles is similar to denervated muscle. We found significant increase in NMJ genes like AChR α , AChR β , Agrin, MuSk and Dok7 in skeletal muscle of $Tak1^{mKO}$ mice compared with $Tak1^{fl/fl}$ mice. We speculate that muscle specific expression of TAK1 maintains the NMJ homeostasis and deletion of TAK1 leads to disruption of normal NMJ morphology and resulting presynaptic and post-synaptic phenotype. Since inactivation of TAK1 also causes oxidative stress in skeletal muscle (Roy et al. FASEB BioAdvances, 2020; 2 (9), 538-553), the disruption of redox homeostasis could be another mechanism for derangement of NMJs. However, further investigations are needed to understand how inactivation of TAK1 in myofibers causes NMJ abnormalities and denervation phenotype.

3. The rationale behind co-administering AAV6-TAK1 with AAV6-TAB1 is explained based on the

literature (Refs 18 and 26). However, I feel it would be particularly informative to present preliminary representative results of experiments performed in mice receiving single AAVs (i.e., AAV6-TAK1 or AAV6-TAB1). This would confirm the assertion that, “TAK1 has no kinase activity when ectopically expressed alone but is activated when co-expressed with TAB1”; this will validate the chosen dual-AAV approach, and confirm that the assertion holds in this experimental setting.

OUR RESPONSE: We thank the reviewer for this important suggestion. Similar experiment was suggested by Reviewer # 2. We have now performed additional experiments in which we also employed AAV6-TAK1 or AAV6-TAB1 alone. As our results in Figure 1 demonstrate that co-injection of AAV-TAK1 and AAV-TAB1, but not AAV-TAK1 or AAV6-TAB1 alone induces myofiber hypertrophy in adult mice. More importantly, our new results clearly demonstrate that the levels of phosphorylated TAK1 are increased only when AAV6-TAK1 was injected along with AAV-TAB1 (Fig. 1g). Furthermore, our new experiments also suggest that co-expression of TAK1 and TAB1, but not TAB1 or TAK1 alone, is essential to activate protein translational machinery in skeletal muscle of mice (Supplemental Fig. S3). Please refer to highlighted text on Page # 6 and 7 for description.

4. Relating to the previous point, TAB1 overexpression upon AAV6-TAB1 treatment is not confirmed at any point and should be.

OUR RESPONSE: We have now shown the expression level of TAB1 by AAV6-TAB1 in our revised manuscript (e.g. Fig 1g and Supplemental Fig. S2d).

5. AAV-mediated expression of GFP and TAB1 are also not confirmed in the cultured myotubes and should be.

OUR RESPONSE: We have now shown the expression of GFP (new Fig. 3a) and TAB1 (Fig. 3b) in cultured myotubes in this revised version of our manuscript.

6. Puromycin levels should really be quantified to confirm the increased translation in the TAK1/TAB1 overexpressing cultures. The ubiquitination experiments in Figure 5 should also be quantified and repeated.

OUR RESPONSE: We have now provided quantification of the levels of puromycin incorporation in cultured myotubes (new Fig. 3e, 3g) and levels of ubiquitinated proteins in TA muscle of mice (now Fig. 6g).

7. Were the immunoprecipitation experiments repeated and/or quantified? Doing so will perhaps allow the authors to see whether the interaction between TAK1 and Smad1 is indeed greater upon denervation, as is hinted at in the text.

OUR RESPONSE: We thank the reviewer for this suggestion. We had repeated the immunoprecipitation experiments thrice. We have now presented the densitometry analysis as bar diagram (new Fig. 6b).

8. In earlier figures (1-4), phosphorylated forms of proteins are quantified relative to the loading control GAPDH; however, in Figures 5 and 8, phosphorylated protein levels are calculated relative to total forms of respective proteins. To aid interpretation, both methods should be applied to all relevant data or perhaps one method used throughout.

OUR RESPONSE: As also suggested by Reviewer # 1, we have made the appropriate changes. We have now separately presented the phospho-proteins and total proteins normalized to loading control (GAPDH or tubulin).

Minor

OUR RESPONSE: Proximal signaling refers to molecules/mechanisms that function at the receptor levels, mostly though interacting with cytoplasmic domain of a receptor. However, TAK1 does not directly interact with the intracellular domains of cell surface receptors. Therefore we have removed the “proximal Signaling” from the Abstract section of our manuscript.

2. The purpose (e.g., skeletal muscle growth) and experimental detail (e.g., muscles removed) of the bilateral synergistic ablation surgery should be better explained on page 5 when first introduced.

OUR RESPONSE: This has now been explained in the “Results” section. Please refer to highlighted text on Page # 5.

3. Line 180: TAB1 should also be included in this final sentence of the section.

OUR RESPONSE: We have rephrased the final sentence as follows: “Collectively, these results suggest that TAK1 is activated during overload-induced muscle growth and forced activation of TAK1 through overexpression of TAK1 and TAB1 is sufficient to promote myofiber hypertrophy in adult mice.” Now on page # 7, highlighted text.

4. The extent of Tak1 knockdown upon tamoxifen treatment should really have been shown in both the tibialis anterior (TA) and gastrocnemius (GA) muscles analyzed, as opposed to just the GA.

OUR RESPONSE: We have confirmed multiple times that tamoxifen treatment causes TAK1 knockdown in all hind limb muscles. We have previously published knockdown in TA muscle in our JCI Insight article (Hindi et al, JCI Insight. 2018 Feb 8;3(3):e98441).

5. To describe the NMJ as having three components, is correct, but only two of the three mentioned are classically described in the tripartite NMJ. The authors should probably mention

that terminal Schwann cells are also a critical player at the neuromuscular synapse if they are going to introduce the NMJ in this way.

OUR RESPONSE: We have now added Schwann cells as well where NMJ are introduced in the “Results” section. Page # 10, highlighted text.

6. Features assessed using NMJ morph need to be better introduced for those not familiar with NMJs and the software. For instance, how do AChR and endplate areas differ? What are compactness and fragmentation?

OUR RESPONSE: We thank the reviewer for this suggestion. We have now included the details of the NMJ feature in the “Methods” section. Please refer to highlighted text on Page # 24.

7. Please include details of how many NMJs per animal were assessed for the morphological analyses – I cannot seem to find the detail.

OUR RESPONSE: We analyzed 20 NMJs per animal. In total, we analyzed 3 floxed and 3 knock out mice for the new experiment. This has been described in “Morphometric analysis of NMJ” part of the “Methods” section and in Figure 4 legends.

8. There appears to be an issue with the NMJ image presented for the control mice (top row, Figure 3A) – the red and blue channels appear shifted to the left and out of sync with the green channel. Also, red/green immunofluorescent colour-combinations should be avoided due to issues for certain forms of colour-blindness – please pseudocolour.

OUR RESPONSE: Per reviewer’s suggestion we have provided new images in pseudocolor.

9. Reference to Fig. 5C is missing from the text (lines 337-339).

OUR RESPONSE: This has been added.

10. The title of Figure S5 contradicts the findings presented in Figure 1E-G.

OUR RESPONSE: This has been corrected. Now supplemental Fig. S8.

Reviewers' Comments:

Reviewer #1:

Remarks to the Author:

The revised version is improved and several of my concerns were addressed. There is only one minor additional point that should be discussed in the manuscript. The authors should discuss their findings on Tak1 involvement in neuromuscular junction maintenance in light of the recent published paper that shows a link between BMPs and myofiber innervation (Sartori R et al. Perturbed BMP signaling and denervation promote muscle wasting in cancer cachexia, *Sci Transl Med* . 2021 Aug 4;13(605):eaay9592.)

Reviewer #2:

Remarks to the Author:

The manuscript has been improved and the reviewer is satisfied with the reply in part. TAK1-TAB story in skeletal muscle biology is interesting.

Some comments should be considered for improvement and acceptance.

In Fig.1 and 5, both Smad1/5/8 and Smad2 is activated. These signals are opposite in regard with muscle homeostasis. Discuss the relationship with the phenotype.

Line 82, growth differentiation factor 15 is proper. Begin with small letters.

Line 92, 'to regulate expression of various molecules' is vague. Define and list specific examples.

Line 120, TGF- β is fine.

Line 122, 'satellite cells' is fine.

Line 249, is $P=0.07$ only for p-Mnk1? Why is P value provided here?

Line 416, Smad6 works in cytoplasm and near the receptors. What are the meanings of nuclear localization of Smad6? Does Smad6 shuttle between nucleus and cytoplasm?

Line 668, add / (slash) between DMEM and F10.

FIG.2, (a)-(d) should be small letters.

FIG.4a and its legend, 2H3 is a name of the clone. Define anti-Neurofilament (NF-M).

In FIG.6, ubiquitination is not so clear.

Reviewer #3:

Remarks to the Author:

The authors have done an excellent job at responding to my comments and those of the other reviewers. The newly presented results allay all of my concerns from the original submission, and I am therefore happy for the manuscript to be published.

RESPONSE TO REVIEWERS' COMMENTS

We sincerely thank all the three reviewers for finding our response adequate and that the revised version of our manuscript much improved. We also thank the reviewers' for providing a few more suggestions. Here is our response to reviewers' comments.

REVIEWER # 1

The revised version is improved and several my concerns were addressed. There is only one minor additional point that should be discussed in the manuscript. The authors should discuss their findings on Tak1 involvement in neuomuscular junction maintenance in light of the recent published paper that shows a link between BMPs and myofiber innervation (Sartori R et al. Perturbed BMP signaling and denervation promote muscle wasting in cancer cachexia, *Sci Transl Med* . 2021 Aug 4;13(605):eaay9592.)

OUR RESPONSE: We thank the reviewer for finding our manuscript improved. The suggested article by the reviewer is highly important. We have now discussed this article in the revised manuscript. Please refer to the second last paragraph in the "Discussion" section of this revised manuscript. Highlighted text on Page # 21.

REVIEWER # 2

The manuscript has been improved and the reviewer is satisfied with the reply in part. TAK1-TAB story in skeletal muscle biology is interesting.

OUR RESPONSE: Thank you for finding our manuscript improved.

Some comments should be considered for improvement and acceptance.

In Fig.1 and 5, both Smad1/5/8 and Smad2 is activated. These signals are opposite in regard with muscle homeostasis. Discuss the relationship with the phenotype.

OUR RESPONSE: We agree with the reviewer that both Smad2/3 and Smad1/5/8 are activated and they are known to have opposing effects on muscle mass. However, both complexes compete for Smad4 for their transcriptional activation. Eventually, the binding of Smad4 with Smad2/3 or Smad1/5/8 dictates which pathway is predominately activated. Under denervated conditions and also during hypertrophy both Smad1/5/8 and Smad2/3 are activated. The reason for activation of Smad2/3 during hypertrophy is unknown. Under denervated condition, we found that TAK1 interacts with Smad1 and influences the spatial regulation of Smads. Future investigations are needed to know whether TAK1 can influence the activity of Smad2/3 in atrophy and hypertrophy. We have now discussed this aspect in the "Discussion" section of our manuscript. Please refer to highlighted text on page # 20.

Line 82, growth differentiation factor 15 is proper. Begin with small letters.

OUR RESPONSE: This has been corrected.

Line 92, 'to regulate expression of various molecules' is vague. Define and list specific examples.

OUR RESPONSE: We have now modified this sentence to include some examples of Smad2/3 target genes such as *Nodal*, *Pitx2*, *Lefty1* and *Lefty2*. Please refer to highlighted text on Page # 3.

Line 120, TGF- β is fine.

OUR RESPONSE: This has been corrected.

Line 122, 'satellite cells' is fine.

OUR RESPONSE: This has been corrected.

Line 249, is P=0.07 only for p-Mnk1? Why is P value provided here?

OUR RESPONSE: We have now removed P value from this bar diagram.

Line 416, Smad6 works in cytoplasm and near the receptors. What are the meanings of nuclear localization of Smad6? Does Smad6 shuttle between nucleus and cytoplasm?

OUR RESPONSE: We agree that Smad6 is predominately present in cytoplasm near the receptor. However, there are several published reports, which demonstrate enrichment of Smad6 in nucleus where it modulates the activity of various activators and repressors of transcription factors (e.g. Bai et al. J Biol Chem. 2000 Mar 24;275(12):8267-70; Jiao et al. Nat Commun. 2018 Jun 27;9(1):2504). We cannot say for sure it is shuttling between nucleus and cytoplasm but our results demonstrate increased levels of Smad6 in nucleus. We have stated in the manuscript that significantly higher levels of Smad6 were observed in the nuclear fraction of denervated muscle.

Line 668, add / (slash) between DMEM and F10.

OUR RESPONSE: This has been added.

FIG.2, (a)-(d) should be small letters.

OUR RESPONSE: This has been corrected.

FIG.4a and its legend, 2H3 is a name of the clone. Define anti-Neurofilament (NF-M).

OUR RESPONSE: This has been corrected. Thank you.

In FIG.6, ubiquitination is not so clear.

OUR RESPONSE: This blot represents total ubiquitination in muscle tissue. We see multiple bands for this type of analysis. We have also now improved the quality of the figure.

REVIEWER # 3

The authors have done an excellent job at responding to my comments and those of the other reviewers. The newly presented results allay all of my concerns from the original submission, and I am therefore happy for the manuscript to be published.

OUR RESPONSE: Thank you for finding our revised manuscript acceptable for publication.